# Homologue replacement in the import motor of the mitochondrial inner membrane of trypanosomes

Corinne von Känel[1], Sergio A Muñoz-Gómez[2], Silke Oeljeklaus[3], Christoph Wenger[1], Bettina Warscheid[3], Jeremy G Wideman[2]*, Anke Harsman[1]*, Andre Schneider[1]*

[1]Department of Chemistry and Biochemistry, University of Bern, Bern, Switzerland; [2]Center for Mechanisms of Evolution, Biodesign Institute, School of Life Sciences, Arizona State University, Tempe, United States; [3]Biochemistry and Functional Proteomics, Institute of Biology II, Faculty of Biology and Signalling Research Centres BIOSS and CIBSS, University of Freiburg, Freiburg, Germany

**Abstract** Many mitochondrial proteins contain N-terminal presequences that direct them to the organelle. The main driving force for their translocation across the inner membrane is provided by the presequence translocase-associated motor (PAM) which contains the J-protein Pam18. Here, we show that in the PAM of *Trypanosoma brucei* the function of Pam18 has been replaced by the non-orthologous euglenozoan-specific J-protein TbPam27. TbPam27 is specifically required for the import of mitochondrial presequence-containing but not for carrier proteins. Similar to yeast Pam18, TbPam27 requires an intact J-domain to function. Surprisingly, *T. brucei* still contains a bona fide Pam18 orthologue that, while essential for normal growth, is not involved in protein import. Thus, during evolution of kinetoplastids, Pam18 has been replaced by TbPam27. We propose that this replacement is linked to the transition from two ancestral and functionally distinct TIM complexes, found in most eukaryotes, to the single bifunctional TIM complex present in trypanosomes.

*For correspondence:
Jeremy.Wideman@asu.edu (JGW);
anke.harsman@web.de (AH);
andre.schneider@dcb.unibe.ch
(AS)

Competing interests: The authors declare that no competing interests exist.

## Introduction

Mitochondria evolved from a single endosymbiotic event between an ancestral alpha-proteobacterium and an archaeon-related host cell. During the gradual conversion of the endosymbiont into an organelle, most of the many genes encoded in the genome of the proto-mitochondrion were either lost or transferred to the host genome (*Roger et al., 2017*; *Archibald, 2015*; *Dacks et al., 2016*). The mitochondrial genes that are now encoded in the nucleus, are translated on cytosolic ribosomes and imported into the various mitochondrial subcompartments. Mitochondrial protein import is executed by several protein complexes, which were thought to be well conserved (*Pfanner et al., 2019*; *Hansen and Herrmann, 2019*). How these translocase complexes originally evolved is important for understanding early eukaryotic evolution specifically, and host-symbiont integration, more generally. However, as mitochondria have a billion-year history and can be traced back to the Last Eukaryotic Common Ancestor (LECA) (*Roger et al., 2017*), it is difficult to gain unambiguous information about their origin and diversification. Although (nearly *Karnkowska et al., 2016*) all eukaryotes have mitochondria or related organelles, recent studies have shown that mitochondria and their import apparatuses are considerably diverse (*Dolezal et al., 2006*; *Harsman and Schneider, 2017*; *Mani et al., 2016*; *Gray, 2012*; *Fukasawa et al., 2017*). Thus, it is important to investigate variation across eukaryotes in order to understand how these absolutely essential import systems have evolved and diversified over the last billion years of eukaryotic evolution.

Mitochondrial protein import has best been studied in *Saccharomyces cerevisiae* and much of this work can, in principle, be generalized to most eukaryotic lineages. However, more recent work has also characterized mitochondrial import pathways in the parasitic protozoan *Trypanosoma brucei*. These investigations have revealed the extreme divergence of the *T. brucei* import machinery compared to other eukaryotes (*Harsman and Schneider, 2017*; *Schneider, 2018a*). The translocase of the outer membrane (TOM) complex in *S. cerevisiae* comprises seven subunits, of which three are conserved across all eukaryotic lineages and can be traced to LECA (*Mani et al., 2016*; *Maćasev et al., 2004*; *Mani et al., 2017*). On the other hand, the atypical translocase of the outer membrane (ATOM) in trypanosomes also consists of seven subunits, but only two of these are shared with all eukaryotes (*Mani et al., 2016*; *Mani et al., 2017*; *Mani et al., 2015*). In the case of the translocase of the inner membrane (TIM) complexes, the situation is even more extreme. Most eukaryotes have two TIM complexes (TIM22 and TIM23) which do not have any subunits in common (*Harsman and Schneider, 2017*; *Žárský and Doležal, 2016*; *Marom et al., 2011*). The TIM22 complex is specialized for membrane insertion of proteins that contain multiple membrane-spanning domains such as mitochondrial carrier proteins (*Ferramosca and Zara, 2013*; *Pfanner and Neupert, 1987*). The TIM23 complex mediates membrane translocation and insertion of presequence-containing proteins (*Pfanner et al., 2019*; *Hansen and Herrmann, 2019*; *Marom et al., 2011*; *Mokranjac and Neupert, 2010*). Surprisingly, trypanosomes have only a single TIM complex that, with minor variations, functions in both import of presequence-containing and carrier proteins (*Harsman et al., 2016*). The only TIM complex component shared between trypanosomes and other eukaryotes is TbTim17 (*Harsman et al., 2016*; *Singha et al., 2012*), an orthologue of the Tim22 subunit of the TIM22 complex (*Žárský and Doležal, 2016*; *Pyrihová et al., 2018*).

How and why did these extreme changes in the trypanosomal TIM complex occur? Is the divergence observed due to some unseen selective pressure, or could it have evolved through neutral evolutionary processes (*Stoltzfus, 1999*; *Lukeš et al., 2011*; *Wideman et al., 2019*)? To start addressing these questions, here we focused on the presequence translocase-associated motor (PAM) of *T. brucei*.

In order to translocate presequence-containing proteins, the TIM23 complex must associate with the matrix-localized PAM, which in yeast, consists of five essential subunits (*Marom et al., 2011*; *Schulz et al., 2015*; *Craig, 2018*). These are (i) the mitochondrial heat shock protein 70 (mHsp70) (*Horst et al., 1997*; *Kang et al., 1990*) and (ii) its nucleotide exchange factor Mge1, which mediate the transport of presequence-containing substrates across the mitochondrial membranes; (iii) Tim44 which recruits mHsp70 to Tim23 (*Banerjee et al., 2015*), the pore-forming subunit of the TIM23 complex.; (iv) Pam18, a J protein with an N-terminal transmembrane domain (TMD) that stimulates ATP hydrolysis of mHsp70 (*D'Silva et al., 2003*; *Truscott et al., 2003*; *Mokranjac et al., 2006*), and is itself regulated by associating with (v) the soluble J-like protein Pam16 (*Li et al., 2004*; *Frazier et al., 2004*). These PAM subunits are relatively well conserved across eukaryotes (*Fukasawa et al., 2017*).

We predict that the single bifunctional *T. brucei* TIM complex needs to associate with a PAM complex to import presequence-containing proteins into the matrix. However, the nature of this trypanosomal PAM is presently unknown. Single trypanosomal orthologues of mHsp70 and Mge1 are readily identified, and an ORF resembling Tim44 (Tb927.7.4620) is detected through HHPred profile-profile analyses. Furthermore, a number of J and J-like proteins are present in the mitochondrial proteome of *T. brucei*, including putative orthologues of Pam 16 and 18 (*Fukasawa et al., 2017*).

Here, we confirm the identity of the Pam16 and Pam18 orthologues in the mitochondrion of *T. brucei* and we demonstrate that they, surprisingly, do not function in the PAM complex. Instead, we show that the J protein TbPam27 is an essential component of the trypanosomal PAM. This suggests that TbPam27 convergently evolved to replace the function of the ancestral Pam18 in kinetoplastids, probably via neutral evolutionary processes.

## Results

### Identification of J domain-containing putative PAM subunits in *T. brucei*

We searched for J domain-containing proteins that could be part of the PAM complex that cooperates with the single bifunctional TIM complex of *T. brucei*. To do so, we first built a protein similarity

network of the J protein superfamily from a set of 46 diverse representative eukaryotes (*Figure 1—figure supplement 2*). We identified four clusters in this network that correspond to putative PAM subunits and selected them for further experimental analyses (*Figure 1A*, *Figure 1—figure supplement 1*). Two clusters correspond to the Pam16 and Pam18 subfamilies of J proteins which are found in diverse eukaryotic representatives, including euglenozoans (e.g., kinetoplastids, diplonemids and euglenids). Moreover, two euglenozoan-specific clusters were of note. One was of special interest since its member in *T. brucei* is a 27 kDa protein (Tb927.10.13830, termed TbPam27) previously found to be associated with the single trypanosomal TIM complex (*Harsman et al., 2016*). Finally, our analysis revealed another kinetoplastid-specific cluster whose member in *T. brucei* is Tb927.4.650 and which appeared closest in similarity to Pam18 and Pam16 in profile Hidden Markov Model (HMM) searches. Thus, our network and phylogenetic analysis (*Figure 1*, *Figure 1—figure supplement 1*, *Figure 1—figure supplement 2*) of J proteins confirms that *T. brucei* has not only orthologues of Pam16 (Tb927.9.13530, termed TbPam16) and Pam18 (Tb927.8.6310, termed TbPam18) (*Fukasawa et al., 2017*) but also kinetoplastid-specific J proteins of which TbPam27 appears to be associated with the TIM complex.

## TbPam27, but not TbPam18, TbPam16 or Tb927.4.650, is associated with the TIM complex

We then tested whether the four J proteins identified through our bioinformatic analyses are localized to mitochondria. Cell extractions with low concentration of digitonin show that the four C-terminally epitope-tagged proteins co-fractionate with the mitochondrial marker ATOM40 (*Figure 2A*). Moreover, the four proteins are all present in the mitochondrial proteome defined by ImportOmics (*Peikert et al., 2017*). All proteins are also exclusively recovered in the pellet when a crude mitochondrial fraction is subjected to carbonate extraction at high pH, suggesting they are integral membrane proteins (*Figure 2B*). Protease protection experiments additionally show that TbPam27, TbPam18 and TbPam16 remain intact in crude mitochondrial fractions and are only digested when the membrane barrier is destroyed by detergent. This indicates that all three tagged proteins are localized inside a membrane-bound compartment (*Figure 2C*). Finally, normalized abundance profiles of TbPam27 over six subcellular fractions, produced in a previous proteomic analysis (*Niemann et al., 2013*), assign it to the mitochondrial inner membrane (IM) (*Figure 2D*).

Immunoprecipitation (IP) experiments, in which tagged TbPam27 is used as the bait, efficiently recovers the TIM subunits TbTim17 and TimRhom I (*Figure 3A*, left most panel). However, the same is not the case when tagged TbPam18, TbPam16 or Tb927.4.650 are used as baits (*Figure 3A*). To analyze the relation of the four candidate proteins to the single trypanosomal TIM complex in more detail, we surveyed the three previously published SILAC (stable isotope labeling by amino acids in cell culture)-IP experiments of TIM subunits (*Harsman et al., 2016*). In these experiments TbTim17, TbTim42 and the small Tim protein TbTim13, a fraction of which is tightly associated with the TIM complex, were used as the baits (*Harsman et al., 2016*; *Wenger et al., 2017*). Moreover, we also performed an analogous SILAC-IP experiment, using the C-terminally triple myc-tagged TIM subunit acyl-CoA dehydrogenase (ACAD) as the bait. The results of all four SILAC-IP experiments are summarized in *Figure 3—figure supplement 1*. The figure depicts the relative enrichment factors for our candidate proteins and for the detected TIM core subunits, compared to the epitope-tagged bait proteins. Interestingly, TbPam27 was efficiently enriched in three out of four experiments. TbPam18, TbPam16 and Tb927.4.650, however, were either not detected in any of the four experiments, or in the case of TbPam16, in the ACAD pull down experiment, not enriched. Thus, TbPam27, but not TbPam18, TbPam16 or Tb927.4.650, is associated with the trypanosomal TIM. This is remarkable, since our protein similarity network shows that TbPam27 is more similar to other members of the J protein superfamily, than to TbPam16 and TbPam18 (*Figure 1*, *Figure 1—figure supplement 1*, *Figure 1—figure supplement 2*). We, therefore, decided to carry out the reciprocal SILAC-IP experiments, using C-terminally HA-tagged TbPam18 and TbPam16 as baits. For TbPam16, the protein that was enriched the most and significantly detected, besides the bait, was TbPam18 (5.7-fold), suggesting that the two proteins interact. Most importantly, however, neither of the eight detected TIM subunits was enriched (*Figure 3B*, left panel; *Figure 3—source data 2*). In the TbPam18-HA SILAC-IP many more proteins were found to be enriched (*Figure 3B*, right panel; *Figure 3—source data 3*). Unexpectedly, essentially all the proteins with the highest significant enrichment are either annotated as ER proteins or predicted to be involved in ER-related processes, such

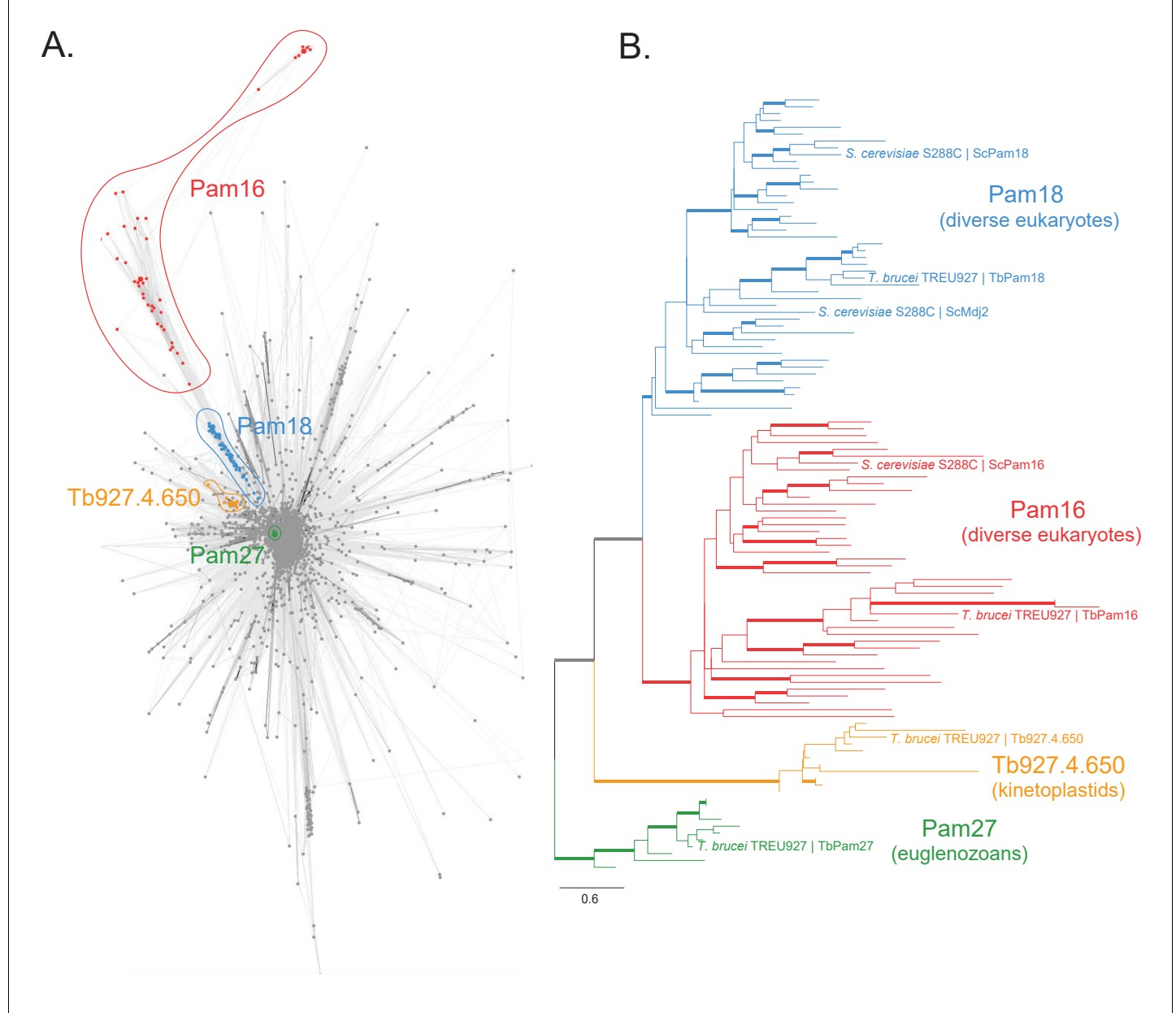

**Figure 1.** Sequence analysis of the J protein superfamily in eukaryotes and euglenozoans. (**A**) Similarity network of DnaJ and Pam16 domain-containing proteins clustered by CLANS with a P-value threshold value of $1 \times 10^{-12}$. A cohesive Pam18 cluster is recovered (blue), as well as a more diffused Pam16 cluster (red). Two smaller and cohesive Kinetoplastea-specific clusters corresponding to Pam27 (green) and Tb927.4.650 (orange) are also recovered. (**B**) Condensed phylogenetic tree inferred by IQ-TREE and the best-fitting LG+R5 model from the Pam16, Pam18, Pam27 and Tb927.4.650 clusters identified in the protein similarity network (see *Figure 1—figure supplement 2* for a fully annotated phylogenetic tree). Thick branches represent branch support values higher than 70% SH-aLRT and 70% UFBoot2+NNI. Only representative protein sequences corresponding to *Saccharomyces cerevisiae* S288C and *Trypanosoma brucei brucei* TREU927 are labeled. The phylogenetic tree recovers clades that correspond to the clusters identified by the protein similarity network. Phylogenetic tree is arbitrarily rooted in the internal branch between the Pam27 and all other clades. - For species name, phylogenetic affiliation and database source for each of the predicted proteomes used in the similarity network and phylogenetic analyses see *Figure 1—source data 1*.

The online version of this article includes the following source data and figure supplement(s) for figure 1:

**Source data 1.** Species name, phylogenetic affiliation and database source for each of the predicted proteomes used in the similarity network and phylogenetic analyses.

**Figure supplement 1.** Similarity network of the J protein superfamiliy clustered with a P-value threshold of $1 \times 10^{-20}$.

**Figure supplement 2.** Fully annotated phylogenetic tree of Pam16, Pam18, TbPam27 and Tb927.4.650 J protein subfamilies.

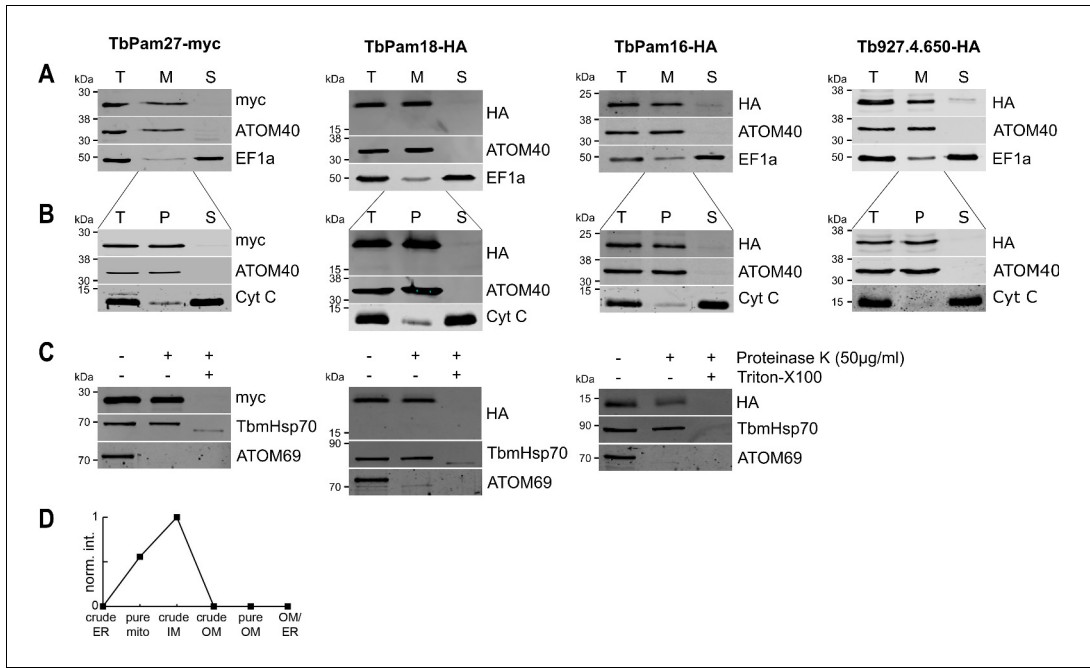

**Figure 2.** TbPam27, TbPam18, TbPam16 and Tb927.4.650 are inner mitochondrial membrane proteins. (**A**) Immunoblot analysis of total cells (T), digitonin-extracted mitochondria-enriched (M) and soluble cytosolic (S) fractions of cells expressing the indicated C-terminally myc- or HA-tagged proteins. The immunoblots were probed with anti-tag antibodies and antisera against ATOM40 and elongation factor 1-alpha (EF1a), which serve as markers for mitochondria and cytosol, respectively. (**B**) Digitonin-extracted mitochondria-enriched fractions (T) were subjected to alkaline carbonate extraction performed at pH 11.5 resulting in membrane-enriched pellet (P) and soluble supernatant (S) fractions. Subsequent immunoblots were decorated with anti-tag antibodies and antisera against ATOM40 and cytochrome C (Cyt C), which serve as markers for integral membrane and soluble proteins, respectively. (**C**) Digitonin-extracted mitochondria-enriched fractions were subjected to a protease protection assay. Subsequent immunoblots were probed with anti-tag antibodies. TbmHsp70 and ATOM69 served as control for matrix and OM proteins, respectively. (**D**) Normalized abundance profile of TbPam27 over six subcellular fractions produced in a previous proteomic analysis (**Niemann et al., 2013**).

as glycosylation or glycosyl-phosphatidyl inositol (GPI) anchoring. Of all the detected mitochondrial proteins, in contrast, only TbPam18 itself was enriched more than 3-fold, and the 11 detected TIM subunits show only a marginal enrichment of 1.6–1.8-fold, considering the 26.3-fold enrichment of the bait. TbPam16 was not detected, which suggests that the C-terminal tag of TbPam18 might prevent the interaction between the two proteins.

In summary, these results confirm that neither TbPam16 nor TbPam18 is associated with the trypanosomal TIM complex. Moreover, based on the SILAC-IP results presented above, it seems that a fraction of TbPam18 might localize to the ER. It is important to note that the SILAC-IPs presented here, were carried out with crude mitochondrial fractions obtained by digitonin extractions (**Figure 2**). In these crude mitochondrial fractions significant amounts of the ER are present. Moreover, both TbPam16 and TbPam18 contain predicted N-terminal mitochondrial targeting sequences. Thus, it is unlikely that the ER localization of TbPam18 is an artefact caused by the C-terminal epitope-tag. Attempts to localize TbPam18 and TbPam16 by immunofluorescence, were not successful, because the signals for TbPam18 and TbPam16 were not strong enough for this type of analysis.

## TbPam27 is part of the active presequence translocase

Using import arrested substrates, it has previously been shown that trypanosomes have a single TIM complex that, with minor variations, mediates import of both presequence-containing and carrier proteins (**Harsman et al., 2016**). The only identified difference between the two forms of the complex was that the two rhomboid-like proteins, TimRhom I and TimRhom II, were specifically associated with the presequence translocase. To investigate whether TbPam27 is associated with the

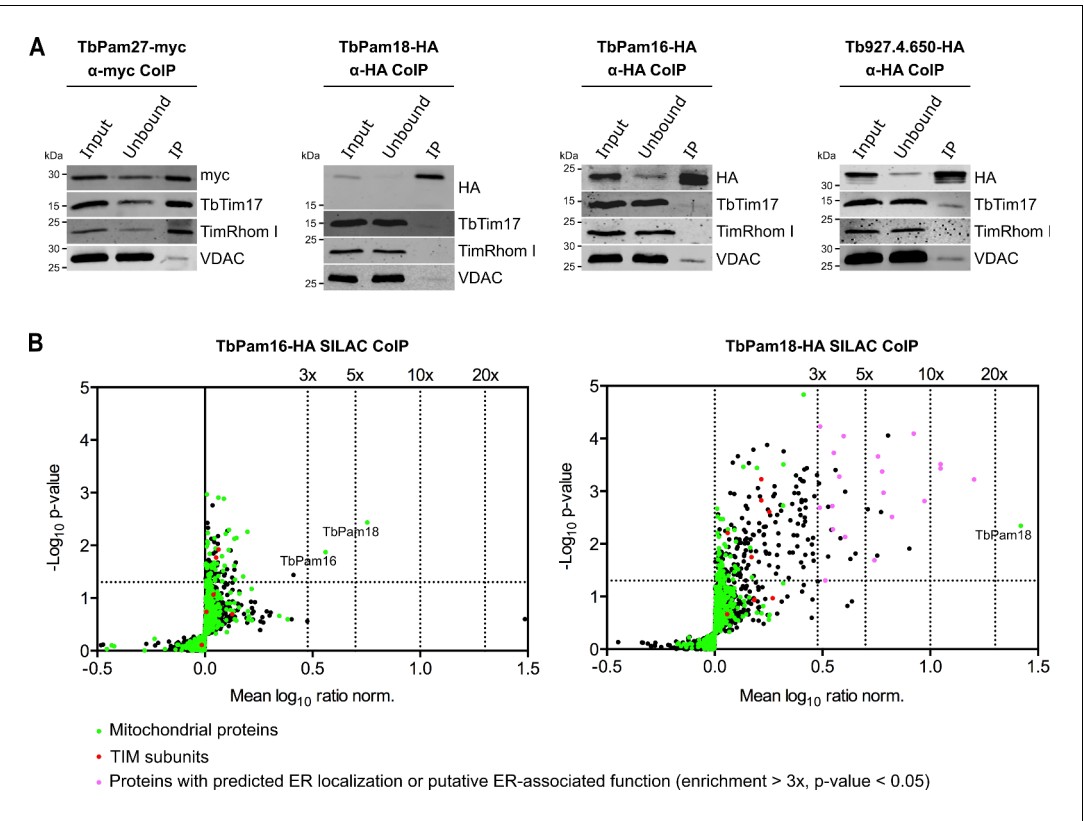

**Figure 3.** TbPam27, but not TbPam18, TbPam16 and Tb927.4.650, is associated with the TIM complex. (**A**) Cell lines expressing the indicated C-terminally triple myc- or HA-tagged proteins were subjected to co-immunoprecipitation (CoIP). 5% each of crude mitochondrial fractions (Input) and unbound proteins (Unbound), as well as 100% of the final eluates (IP) were separated by SDS-PAGE. The resulting immunoblots were probed with anti-tag antibodies and antisera against TIM subunits (TbTim17 and TimRhom I) and VDAC. (**B**) SILAC-IP experiments of TbPam16-HA and TbPam18-HA from digitonin-extracted mitochondria enriched fractions. Volcano plots show the mean $\log_{10}$ ratios (induced/uninduced) of proteins that were detected in at least two of three independent biological replicates by quantitative MS, plotted against the corresponding -$\log_{10}$ P values (one-sided t-test). The vertical dotted lines specify the indicated enrichment factors. The horizontal dotted line indicates a t-test significance level of 0.05. Green dots represent proteins that are found in the mitochondrial proteome (*Peikert et al., 2017*), red dots indicate TIM subunits and pink dots represent proteins that are enriched more than three-fold and which, according to the TriTryp database (https://tritrypdb.org/tritrypdb/), are ER-localized or have a putative ER-associated function.

The online version of this article includes the following source data and figure supplement(s) for figure 3:

**Source data 1.** List of proteins identified by SILAC-MS from a pull down experiment of mitochondria-enriched fractions using C-terminally myc-tagged ACAD as a bait.

**Source data 2.** List of proteins identified by SILAC-MS from a pull down experiment of mitochondria-enriched fractions using C-terminally HA-tagged TbPam16 as a bait.

**Source data 3.** List of proteins identified by SILAC-MS from a pull down experiment of mitochondria-enriched fractions using C-terminally HA-tagged TbPam18 as a bait.

**Figure supplement 1.** Global proteomic analysis confirm that TbPam27, but not TbPam18, TbPam16 or Tb927.4.650, is associated with the TIM complex.

active presequence translocase, we co-expressed TbPam27-myc with a fusion protein. This fusion protein is composed of the presequence and part of the mature form (1–160 aa) of the trypanosomal matrix protein dihydrolipoyl dehydrogenase (LDH), and mouse dihydrofolate reductase (DHFR) carrying a C-terminal triple HA-tag. Addition of aminopterin to this cell line irreversibly fixes the DHFR in its 3D conformation, causing the LDH part of the substrate to be stuck in both the ATOM and the TIM complex (*Harsman et al., 2016*; *Schneider, 2018b*). *Figure 4A* shows that CoIP experiments using the stuck substrate in the presence of aminopterin as the bait, not only recover the LDH-

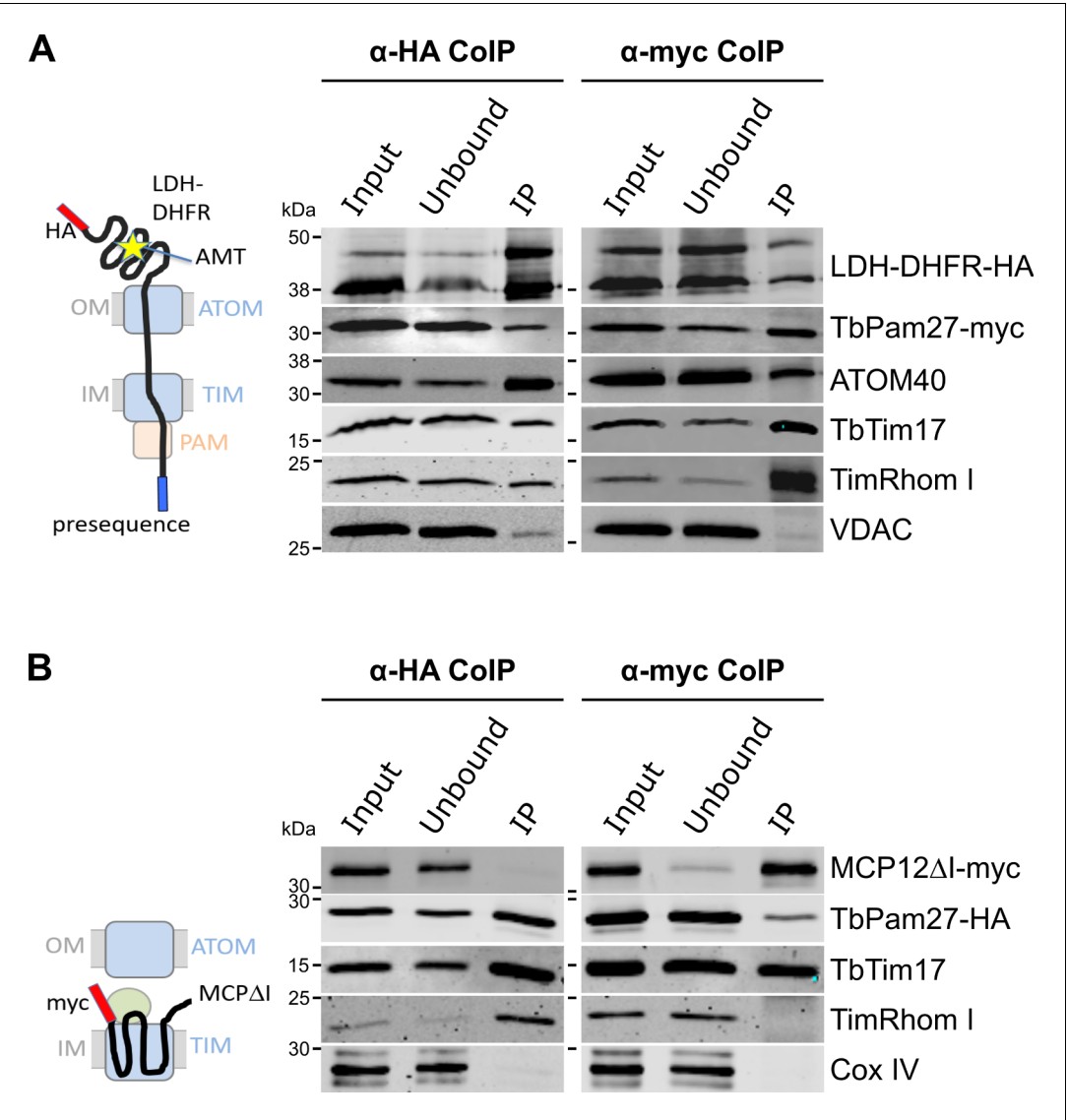

**Figure 4.** TbPam27 is part of the active presequence translocase but not of the active carrier translocase. (**A**) Left: Schematic representation of the stalled presequence intermediate induced by in vivo expression of the LDH-DHFR fusion protein in presence of aminopterin (AMT). Right: Digitonin-solubilized mitochondrial extracts from cells co-expressing LDH-DHFR-HA and TbPam27-myc which were grown in presence of AMT and subjected to co-immunoprecipitation (CoIP) using either the HA- (left panel) or myc-tag (right panel). (**B**) Left: Schematic depiction of the stalled carrier intermediate induced by the expression of a truncated mitochondrial carrier protein (MCPΔI). Right: Digitonin-solubilized mitochondrial extracts from cells co-expressing MCP12ΔI-myc and TbPam27-HA were subjected to CoIP using either the HA- (left panel) or myc-tag (right-panel). For all CoIPs 5% each of crude mitochondrial fractions (Input) and unbound proteins (Unbound), as well as 100% of the final eluate (IP) were separated by SDS-PAGE. The resulting immunoblots were probed with anti-tag antibodies and antisera against ATOM40, TIM subunits (TbTim17 and TimRhom I), VDAC and Cox IV as indicated. The two bands shown for LDH-DHFR-HA, represent the LDH-DHFR-HA portions without (lower band) or with bound AMT (higher band).

DHFR-HA along with ATOM40 and the TIM components TbTim17 and TimRhom I, as expected, but also the myc-tagged TbPam27. Moreover, the reverse experiment, using TbPam27-myc as the bait, precipitates the very same proteins LDH-DHFR-HA, ATOM40, TbTim17 and TimRhom I. However, only negligible amounts of the OM protein VDAC, which was used as a control, were recovered in either of the two experiments.

To investigate whether TbPam27 is also part of the active carrier translocase, we co-expressed C-terminally triple HA-tagged TbPam27 (TbPam27-HA) with a truncated version of the C-terminally myc-tagged carrier protein MCP12 (MCP12ΔI-myc). This substrate was previously shown to accumulate in the TIM complex, engaged in carrier protein import, because it lacks the two N-terminal TMDs (*Harsman et al., 2016*; *Brandner et al., 2005*). IP experiments in *Figure 4B* using anti-myc antibodies, show that TbTim17 was efficiently co-precipitated with MCP12ΔI-myc, but that only marginal amounts of TbPam27-HA were recovered in the eluate. Moreover, as expected, the presequence translocase-specific TimRhom I, as well as cytochrome oxidase subunit 4 (Cox IV), remained in the unbound fraction. Conversely, IP using TbPam27-HA efficiently precipitated TbTim17 and TimRhom I, whereas MCP12ΔI-myc and Cox IV were not recovered.

The results discussed above, are in line with data obtained in a previous study, which analyzed CoIP experiments using the stuck LDH-DHFR precursor or the import-arrested truncated carrier protein as baits by SILAC proteomics (*Harsman et al., 2016*). *Figure 3—figure supplement 1* depicts the obtained enrichment factors for TbPam27, TbPam18 and TbPam16 as well as for the indicated TIM subunits relative to the TIM core subunit TbTim17. The figure illustrates that TbPam27 is much higher enriched in the presequence rather than the carrier translocase SILAC-IP. Moreover, TbPam16 and TbPam18 are either not detected in these experiments or enriched to a lesser extent than the control protein VDAC.

In summary, these experiments show that TbPam27, but neither TbPam16 nor TbPam18, is specifically associated with the active presequence translocase.

## TbPam27 with an intact J domain is essential for mitochondrial protein import

To investigate the function of TbPam27, we produced an inducible RNAi cell line. *Figure 5A*, left panel shows that ablation of TbPam27 causes a rapid growth arrest and a concomitant accumulation of unprocessed Cox IV. Importantly, the levels of TbTim17, one of the core subunits of the trypanosomal TIM complex, are not affected. Previous work has shown that accumulation of precursor Cox IV indicates that mitochondrial import of presequence-containing proteins is inhibited (*Peikert et al., 2017*; *Pusnik et al., 2011*). Essentially identical results as for TbPam27 RNAi were obtained when the analogous experiment was done for the single trypanosomal mitochondrial heat shock protein 70 (TbmHsp70). TbmHsp70, as shown before, is required for matrix protein import (*Tschopp et al., 2011*; *Figure 5A*, right panels). These results show that TbPam27 and TbmHsp70 are essential for growth and that their function is linked to mitochondrial protein import.

Trypanosomes have a complex life cycle alternating between an insect vector, the Tsetse fly, and a mammalian host. The insect-stage procyclic form of the parasite has highly active mitochondria capable of oxidative phosphorylation. The mitochondrion of the bloodstream form, on the other hand, lacks the respiratory complexes, except for the ATP-synthase which functions in reverse (*Schnaufer et al., 2002*). Mitochondrial protein import, however, is essential for both the insect stage and bloodstream form of trypanosomes (*Cristodero et al., 2010*). Thus, should TbPam27 indeed be required for protein import, it will be essential in the bloodstream form as well. We tested this using an inducible TbPam27 RNAi cell line of the *T. brucei* bloodstream strain γL262P (*Dean et al., 2013*). The result in *Figure 5C* shows that TbPam27, indeed, is also essential for normal growth of *T. brucei* bloodstream forms.

In yeast and humans, Pam18, but not Pam16, requires an intact J domain for its function (*D'Silva et al., 2003*; *Davey et al., 2006*; *Sinha et al., 2016*). To study whether this is also the case for TbPam27, we produced an RNAi cell line targeting the 5' untranslated region (UTR) of the TbPam27 mRNA. Such a cell line can be used for complementation experiments provided the TbPam27 open reading frame (ORF) is expressed in the context of a different 5' UTR. The left panels in *Figure 5B* show that, as expected, ectopic expression of untagged TbPam27 in the 5' UTR RNAi cell line restores growth to nearly wild-type levels and prevents Cox IV precursor accumulation. However, when the same experiment was performed with a variant of TbPam27 containing a single H to Q substitution at position 77 in the conserved HPD motif of the J domain, no complementation was observed (*Figure 5B*, right panels). Cell growth was arrested and accumulation of Cox IV precursor, just as the TbPam27 RNAi cell line alone, was observed. The Northern blots in *Figure 5AB* confirm that endogenous TbPam27 mRNA is efficiently down regulated in all RNAi cell lines. The lower

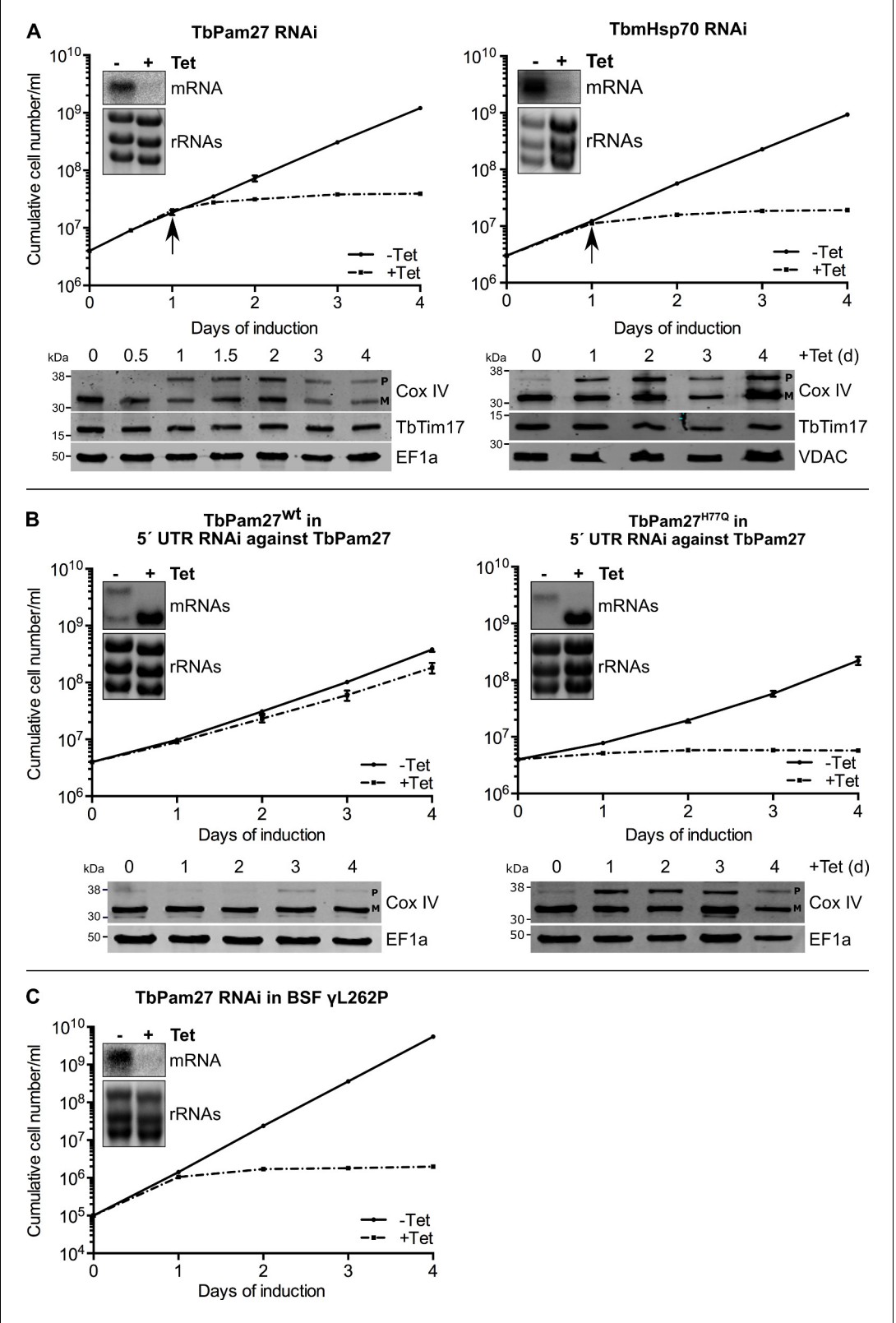

**Figure 5.** TbPam27 is essential and requires an intact J domain for its function. (**A**) Top panel: Growth curves of uninduced (-Tet) and induced (+Tet) procyclic TbPam27 and TbmHsp70 RNAi cell lines. Error bars correspond to the standard deviation (n = 3). Insets show Northern blots of total RNA from uninduced or two days induced cells which were probed for the corresponding mRNAs. Ethidiumbromide-stained rRNAs serve as loading controls.

*Figure 5 continued on next page*

*Figure 5 continued*

Arrows indicate the onset of the protein import phenotype. Bottom panels: Immunoblot analysis of steady-state protein levels of Cox IV and TbTim17 in whole-cell extracts of the respective RNAi cell line. (B) Top panel: Growth curves of uninduced (-Tet) and induced (+Tet) TbPam27 RNAi cell lines targeting the 5' UTR that ectopically either express wildtype TbPam27 (TbPam27$^{wt}$) or a version of TbPam27 carrying the H77Q mutation (TbPam27$^{H77Q}$). Insets show Northern blots of total RNA from uninduced and two days induced cells, probed for the TbPam27 ORF. The top bands correspond to the endogenous and the bottom bands to the ectopically expressed TbPam27 versions. Ethidiumbromide-stained rRNAs serve as loading controls. Bottom panel: Immunoblot analysis of steady-state protein levels of Cox IV in whole-cell extracts of the respective cell line. In (A) and (B) EF1a or VDAC serve as loading control. The positions of Cox IV precursor (P) and mature (M) forms are indicated. (C) Growth curve of the uninduced (-Tet) and induced (+Tet) bloodstream form (BSF) γL262P RNAi cell line ablating TbPam27. Error bars correspond to the standard deviation (n = 3). Insets show Northern blots of total RNA from uninduced or two days induced cells which were probed for the corresponding mRNAs. Ethidiumbromide-stained rRNAs serve as loading controls.

bands in the Northern blots shown in *Figure 5B* furthermore demonstrate that similar amounts of the ectopically expressed TbPam27 mRNAs are produced in the two complemented cell lines.

Thus, TbPam27, as yeast and human Pam18, but unlike Pam16, requires an intact J domain to exert its function.

## Neither TbPam16 nor TbPam18 is required for mitochondrial protein import

We also produced inducible RNAi cell lines of procyclic *T. brucei* for TbPam18, TbPam16 and Tb927.4.650. *Figure 6A* shows that ablation of TbPam18 and TbPam16 causes a growth arrest and an accumulation of unprocessed Cox IV. However, unlike for TbPam27, precursor accumulation was only observed two days after the growth phenotype became apparent, suggesting that it is an indirect effect. Ablation of Tb927.4.650, on the other hand, affected cell-growth only slightly and caused no Cox IV precursor accumulation.

As discussed above, TbPam18 and TbPam16 are not associated with the TIM complex. This result strongly suggests that they are not involved in mitochondrial protein import. To further analyze the functions of TbPam18 and TbPam16, we tested whether the two proteins are essential for normal growth of the bloodstream form γL262P cell line. The results in *Figure 6B* show that, in contrast to what was seen for TbPam27 (*Figure 5C*), this is not the case. This result indicates that neither TbPam18 nor TbPam16 have an essential function in mitochondrial protein import. Thus, while we do not know what the essential functions of TbPam16 and TbPam18 are, they are limited to the procyclic form.

TbPam16 and TbPam18 are not associated with the TIM complex and not involved in protein import. Could it be that the two proteins function as a backup system in the absence of TbPam27? In order to test whether TbPam16 or TbPam18 are recruited to TIM complex in the absence TbPam27, we expressed tagged versions of each of the proteins in the TbPam27 RNAi cell line. The results in *Figure 6—figure supplement 1A* show that neither TbPam16 nor TbPam18 accumulate in the absence of TbPam27. Moreover, CoIP experiments, in which TbPam18-HA or TbPam16-HA in TbPam27 depleted cells were used as baits, still do not recover the TIM subunits TbTim17 or TimRhom I (*Figure 6—figure supplement 1B*). Thus, TbPam18 and TbPam16 are not associated with the TIM complex, irrespectively whether TbPam27 is present or not.

## TbPam27 is required for the formation of the presequence pathway intermediate

Formation of the stalled presequence pathway intermediate (*Figure 4A*) in the presence of aminopterin is expected to depend on an active PAM module (*Schulz and Rehling, 2014*), which pulls the unfolded moiety of the LDH-DHFR-myc precursor across the ATOM and the TIM complexes into the matrix. Formation of the carrier intermediate, however, should be independent of the PAM module (*Chacinska et al., 2009*). Blue native (BN)-PAGE analysis demonstrates that LDH-DHFR-myc accumulates in a high molecular weight complex corresponding to the import intermediate that is stuck in the import complexes as shown in the left panel of *Figure 7A*. However, complex formation is

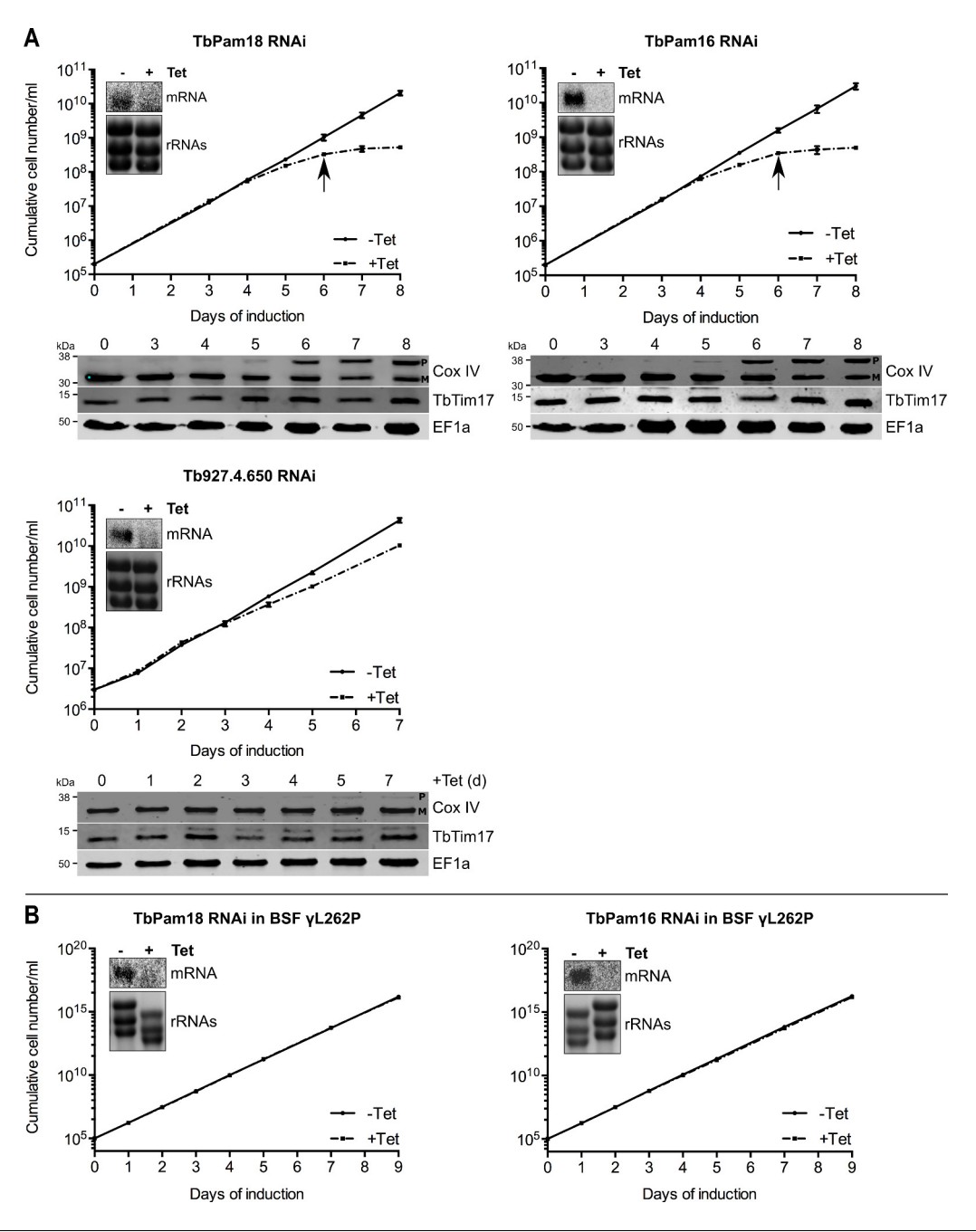

**Figure 6.** Ablation of TbPam18 and TbPam16 affects growth of procyclic form but not of bloodstream form trypanosomes. (**A**) Upper panels: Growth curves of uninduced (-Tet) and induced (+Tet) procyclic TbPam18, TbPam16 and Tb927.4.650 RNAi cell lines. Arrows indicate the onset of the mitochondrial protein import phenotype. Bottom panels: Immunoblot analysis of steady-state protein levels of Cox IV and TbTim17 in whole-cell extracts of the respective RNAi cell lines. EF1a serves as loading control. The positions of Cox IV precursor (P) and mature (M) forms are indicated. (**B**) Growth curve of the uninduced (-Tet) and induced (+Tet) blood stream form (BSF) γL262P RNAi cell lines ablating TbPam18 or TbPam16. In (**A**) and (**B**) error bars correspond to the standard deviation (n = 3). Insets show Northern blots of total RNA from uninduced or two days induced cells which were probed for the corresponding mRNAs. Ethidiumbromide-stained rRNAs serve as loading controls.

The online version of this article includes the following figure supplement(s) for figure 6:

**Figure supplement 1.** TbPam18 and TbPam16 are not associated with the TIM complex irrespectively whether TbPam27 is present or not.

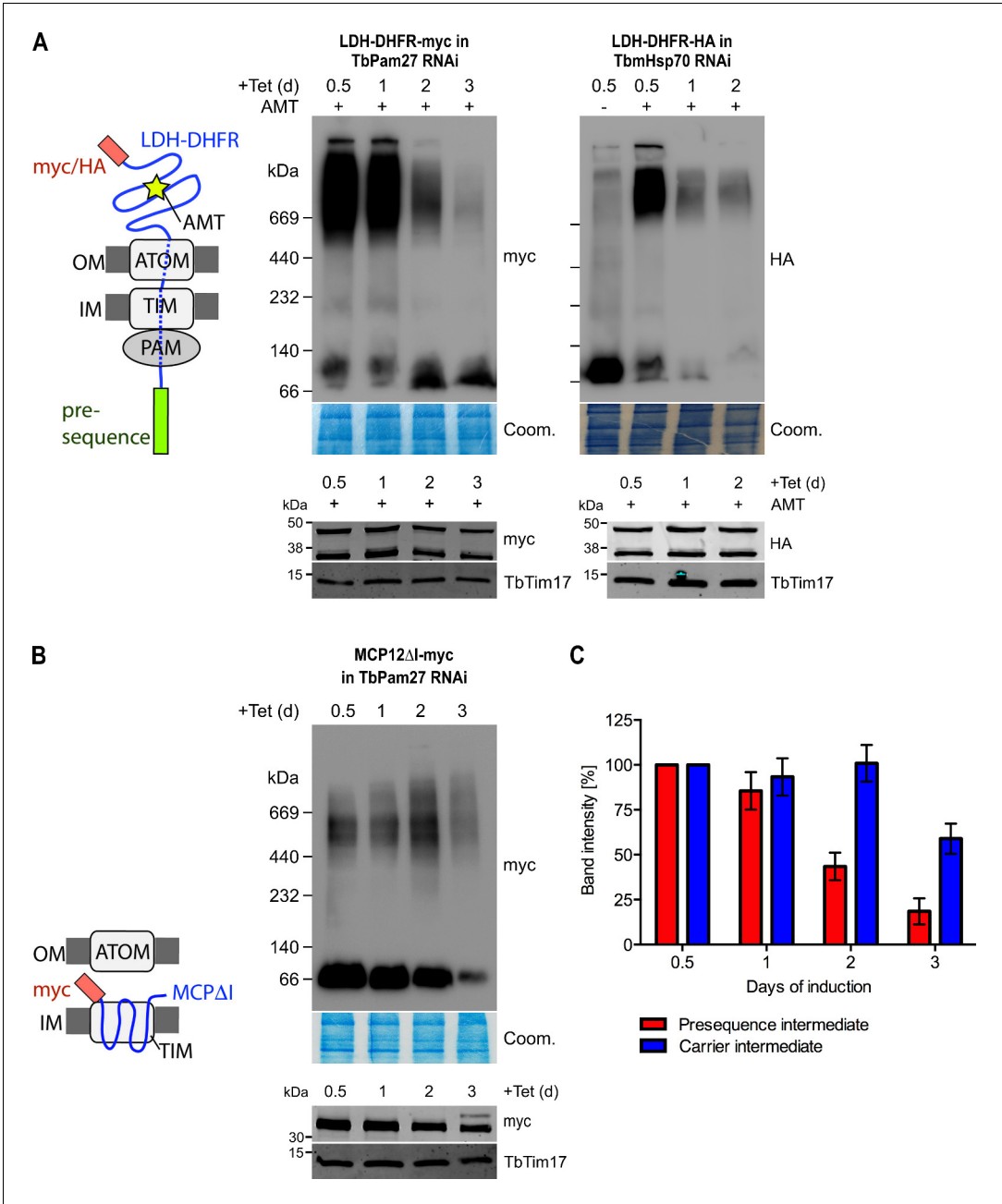

**Figure 7.** TbPam27 is required for the formation of the presequence but not for the carrier intermediate. (**A**) Left: Schematic representation of the stalled presequence intermediate induced by in vivo expression of the LDH-DHFR fusion protein in presence of aminopterin (AMT). Right, upper panels: BN-PAGE analysis of the presequence intermediate in cell lines expressing HA- or myc-tagged LDH-DHFR in the background of RNAi against either TbPam27 or TbmHsp70 as indicated. Cells were grown in the absence or presence (0.5 days) of AMT as specified. (**B**) Left: Schematic depiction of the stalled carrier intermediate induced by the expression of the truncated mitochondrial carrier protein (MCPΔI). Right, upper panel: BN-PAGE analysis of the carrier intermediate in a cell line expressing myc-tagged MCP12ΔI in the background of RNAi against TbPam27. For all experiments digitonin-extracted mitochondrial fractions were prepared after the indicated number of days of RNAi induction and separated on a 4–13% BN-PAGE. The resultant immunoblots were probed with anti-tag antibodies. Coomassie-stained gel sections (Coom.) serve as loading controls. In (**A**) and (**B**) the right, lower panels show SDS-PAGE analysis of whole-cell extracts of the respective cell lines. Immunoblots were probed with anti-tag antibodies, to demonstrate a constant LDH-DHFR-myc/-HA or MCP12ΔI-myc expression. The two bands shown for LDH-DHFR-myc/-HA, represent the LDH-DHFR-myc/-HA portions without (lower band) or with bound AMT (higher band).

*Figure 7 continued on next page*

*Figure 7 continued*
Probing for TbTim17 demonstrates integrity of the TIM complex. (C) Densitometric quantification of the BN-PAGE signals to compare the amounts of LDH-DHFR-myc (as shown in A) and MCP12ΔI-myc (as shown in B) in the TbPam27-RNAi cell line found in the respective high molecular weight complexes. The levels after 0.5 days of RNAi induction were set to 100%. Error bars correspond to the standard error of the mean of three biological replicates.

largely abolished in induced TbPam27 RNAi cells after the time point, when the arrest in growth becomes apparent. As a positive control, the analogous experiment was done in a TbmHsp70 RNAi cell line, and essentially the same result was obtained (*Figure 7A*, right panel). These results demonstrate that both, TbPam27 and TbmHsp70, are required for the formation of the stalled presequence pathway intermediate.

Interestingly, accumulation of the truncated MCP12ΔI-myc at the TIM complex was not affected by ablation of TbPam27 for at least 2 days (*Figure 7B*). Thus, as expected for a PAM subunit, ablation of TbPam27 selectively interferes with the formation of the stalled presequence intermediate but not with formation of the carrier intermediate (*Figure 7C*).

Stalling of the presequence pathway intermediate was also analyzed in the TbPam18 and TbPam16 cell lines, whose ablation caused Cox IV precursor accumulation albeit at late time points only (*Figure 6*). *Figure 8* shows that ablation of neither of the two proteins interferes with the formation of the import arrested presequence-containing substrate.

## Ablation of TbPam27 preferentially affects presequence-containing proteins

To determine the import of which substrate proteins is affected by TbPam27 ablation, we performed a quantitative proteomic analysis of the steady-state levels of mitochondrial proteins in the TbPam27 RNAi cell line. To do so, we used SILAC combined with high-resolution mass spectrometry (MS). Uninduced and induced TbPam27 RNAi cells were grown in medium containing different stable isotope-labelled forms of arginine and lysine. After 1.5 days of RNAi induction equal cell numbers of

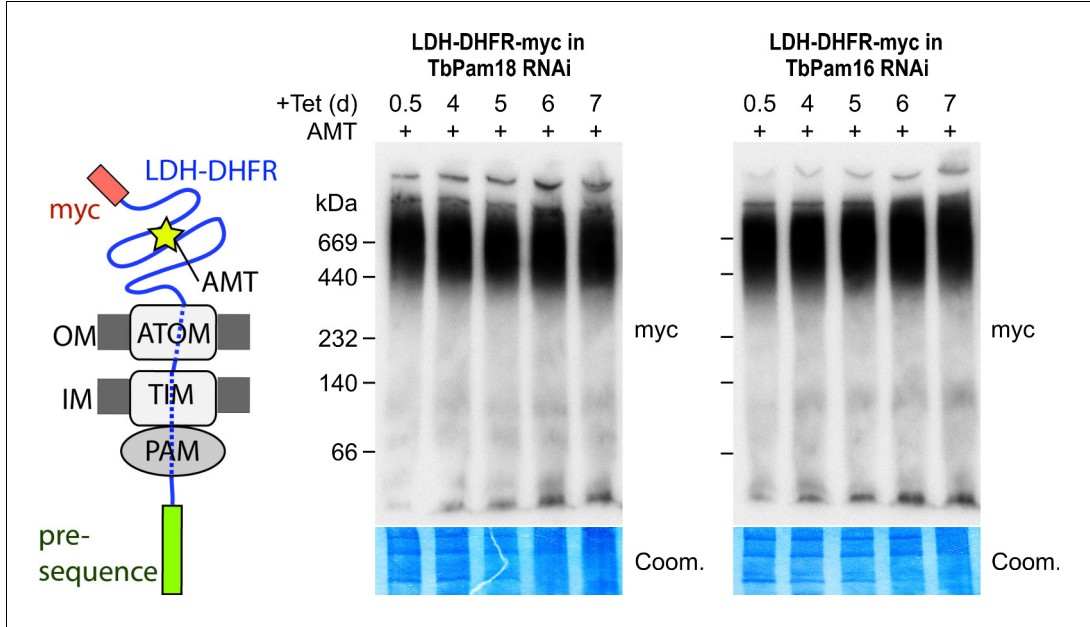

**Figure 8.** Ablation of neither TbPam18 nor TbPam16 does affect the formation of the presequence intermediate. Left: Schematic representation of the stalled presequence intermediate induced by in vivo expression of the LDH-DHFR fusion protein in presence of aminopterin (AMT). Right: BN-PAGE analysis of the presequence intermediate in cell lines expressing myc-tagged LDH-DHFR in the background of RNAi against either TbPam18 or TbPam16 as indicated. Experiments were done exactly as described in *Figure 7A*.

uninduced and induced cultures were mixed and mitochondria-enriched fractions were prepared for further analysis by quantitative MS. At this time point, growth of the induced cells is essentially identical to the uninduced ones (*Figure 5A*). The proteins detected in this analysis, were filtered for mitochondrial proteins, using the recently published mitochondrial proteome of *T. brucei* (*Peikert et al., 2017*). It has previously been shown that a reduction in the abundance of mitochondrial proteins in these types of experiments, is mainly due to inhibition of mitochondrial protein import (*Peikert et al., 2017*; *Pusnik et al., 2011*). We detected 899 mitochondrial proteins, which corresponds to 83% of the previously determined proteome (*Peikert et al., 2017*), and of which 34.4% are downregulated more the 1.5-fold. (*Figure 9A*). Of all detected mitochondrial proteins, 66.5% contain predicted mitochondrial presequences (*Figure 9B*). 39.6% of the presequence-containing proteins show more than 1.5-fold decreased steady-state levels 1.5 days after TbPam27 RNAi induction. However, only 24.6% of the remaining mitochondrial proteins, that presumably have internal targeting sequences (excluding the 12 detected carrier proteins), are reduced more than 1.5-fold (*Figure 9C*). Finally, looking specifically at the mitochondrial carrier proteins, we found that only one of them is more than 1.5-fold downregulated (8.3%) (*Figure 9D*).

In summary, these experiments show that ablation of TbPam27, as expected for a PAM subunit, preferentially affects the presequence pathway. The results also demonstrate that ablation of TbPam27 does not affect the levels of TIM complex subunits (*Figure 9—figure supplement 1*), indicating that the trypanosomal TIM complex and the putative PAM complex are separate entities.

## Discussion

In contrast to all other major eukaryotic lineages, kinetoplastids have a single bifunctional TIM complex which facilitates the import of both, mitochondrial presequence-containing and mitochondrial carrier proteins. Here, we show that TbPam27 is an essential mitochondrial IM protein. It is associated with the presequence translocase but not with the carrier translocase version of the trypanosomal TIM and functions analogously to fungal Pam18. Like Pam18, TbPam27 is an integral IM protein and requires an intact J domain for its function. However, unlike Pam18 subfamily members, which have N-terminal TMDs, TbPam27 is C-terminally anchored in the IM membrane. Consequently, the J domain of TbPam27 has a reverse orientation relative to the IM when compared to Pam18 of yeast and humans (*Mokranjac et al., 2006*). It is unclear if this topological reorientation is functionally significant.

PAM has been studied in-depth exclusively in fungi and animals. There are some data which show that *Trichomonas* and *Giardia* Pam18s localize to hydrogenosomes and mitosomes, respectively (*Rada et al., 2011*; *Dolezal et al., 2005*). A rudimentary functional conservation has been demonstrated as the J domain of the *Trichomonas* Pam18 can complement the J domain of *S. cerevisiae* Pam18 (*Rada et al., 2011*). The PAM subunit Tim44 has orthologues in essentially all eukaryotes (*Clements et al., 2009*) including *Trichomonas* and *Giardia* (*Pyrihová et al., 2018*; *Rada et al., 2011*; *Martincová et al., 2015*). Trypanosomes, on the other hand, lack a clear Tim44 orthologue (*Pyrihová et al., 2018*; *Martincová et al., 2015*). A protein, whose C-terminal domain shows similarity to Tim44 by HHPred has been identified in *T. brucei*. However, it has been shown that this protein is not associated with the trypanosomal TIM complex (*Harsman et al., 2016*). These data suggest that *Trichomonas* and *Giardia* may have a somewhat conventional PAM complex. The TIM-PAM complex in *T. brucei* is thus an outlier. Therefore, it constitutes an excellent system to investigate how a process can be evolutionarily conserved in the midst of sequence divergence and component replacement (*Wideman et al., 2019*).

Surprisingly, *T. brucei* retains orthologues of Pam16 and Pam 18 in its mitochondrion. However, we show that neither of the two proteins is associated with the TIM complex, nor does their ablation interfere with the formation of the presequence pathway intermediate. Furthermore, unlike TbPam27, they are not essential for the bloodstream form of *T. brucei*. These data indicate that they are not subunits of the trypanosomal PAM. Instead, the ancestral function of Pam18 has likely been taken over by TbPam27. Previous examples of convergent evolution for the mitochondrial protein import system of *T. brucei* have been described. They include the two OM import receptors ATOM46 and ATOM69 (*Mani et al., 2015*) as well as pATOM36 (*Vitali et al., 2018*). In these cases, functionally similar proteins convergently evolved likely due to similar selective pressures.

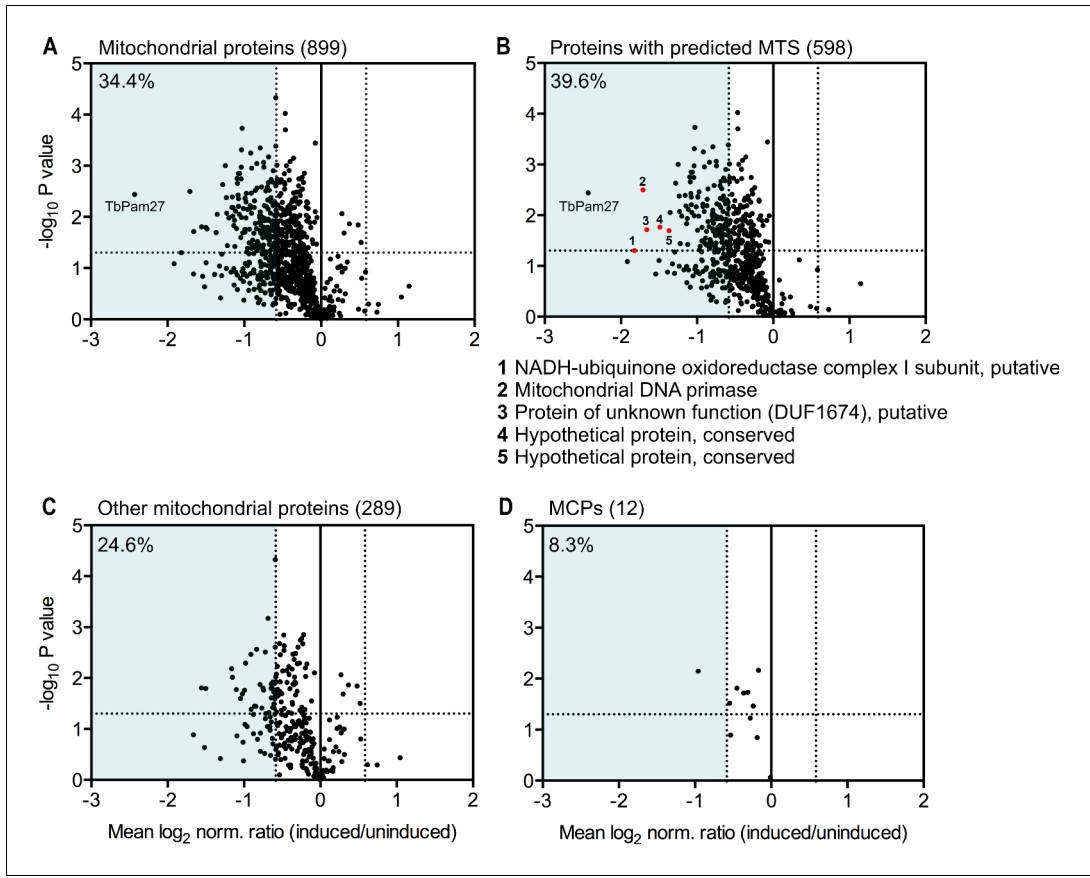

**Figure 9.** Global mitochondrial proteome changes upon ablation of TbPam27. Digitonin-extracted mitochondria-enriched fractions of uninduced and 1.5 days induced TbPam27 RNAi cells were subjected to SILAC-based quantitative MS. The mean $\log_2$ of normalized ratios (induced/uninduced) was plotted against the corresponding -$\log_{10}$ P value (two sided t-test). (**A**) Volcano plot depicts all mitochondrial proteins identified in this experiment (*Peikert et al., 2017*). (**B**) Volcano plot shows proteins with a predicted mitochondrial targeting sequence (MTS) (*Almagro Armenteros et al., 2019*). Numbered red dots and the respective protein names below the figure, correspond to the five most downregulated, significantly detected proteins besides TbPam27. (**C**) Volcano blot depicts the remaining other mitochondrial proteins, which were detected in this experiment, besides mitochondrial carrier proteins (MCPs), which are shown in (**D**). Numbers in parentheses specify how many proteins were detected in each subset. The horizontal dotted line indicates a t-test significance level of 0.05. The vertical dotted lines mark a fold-change in protein abundance of ±1.5 in each plot. The blue background indicates a downregulation of 1.5-fold. The percentage of proteins that are downregulated more than 1.5-fold in each sub-dataset is indicated in the upper left corner of each panel. For a list of all proteins identified in the SILAC-MS experiment see *Figure 9—source data 1*.

The online version of this article includes the following source data and figure supplement(s) for figure 9:

**Source data 1.** List of proteins identified in SILAC-MS experiments of mitochondria-enriched fractions from TbPam27 RNAi cells.

**Figure supplement 1.** Ablation of TbPam27 has no direct effect on the TIM complex stability.

In the case of the TIM complex of kinetoplastids, we have a different story. Previous work has shown that LECA most likely already had both, TIM23 and TIM22 complexes (*Fukasawa et al., 2017*; *Žárský and Doležal, 2016*; *Pyrihová et al., 2018*). Thus, how an ancestor with two specialized import systems evolved to have a single bifunctional TIM complex is a curious affair. We suggest that a neutral replacement of the ancestral Pam18, and possibly also Pam16, by TbPam27 took place in an ancestor of the kinetoplastids, or even possibly at the dawn of the Glycomonada (Kinetoplastea and Diplonemea), as an orthologue of TbPam27 is found in *Diplonema sp.* (*Cavalier-Smith, 2016*). This replacement could explain the transition to a single TIM complex (*Figure 10*). The core subunits of the ancestral TIM22 and TIM23-PAM complexes were likely very similar to the

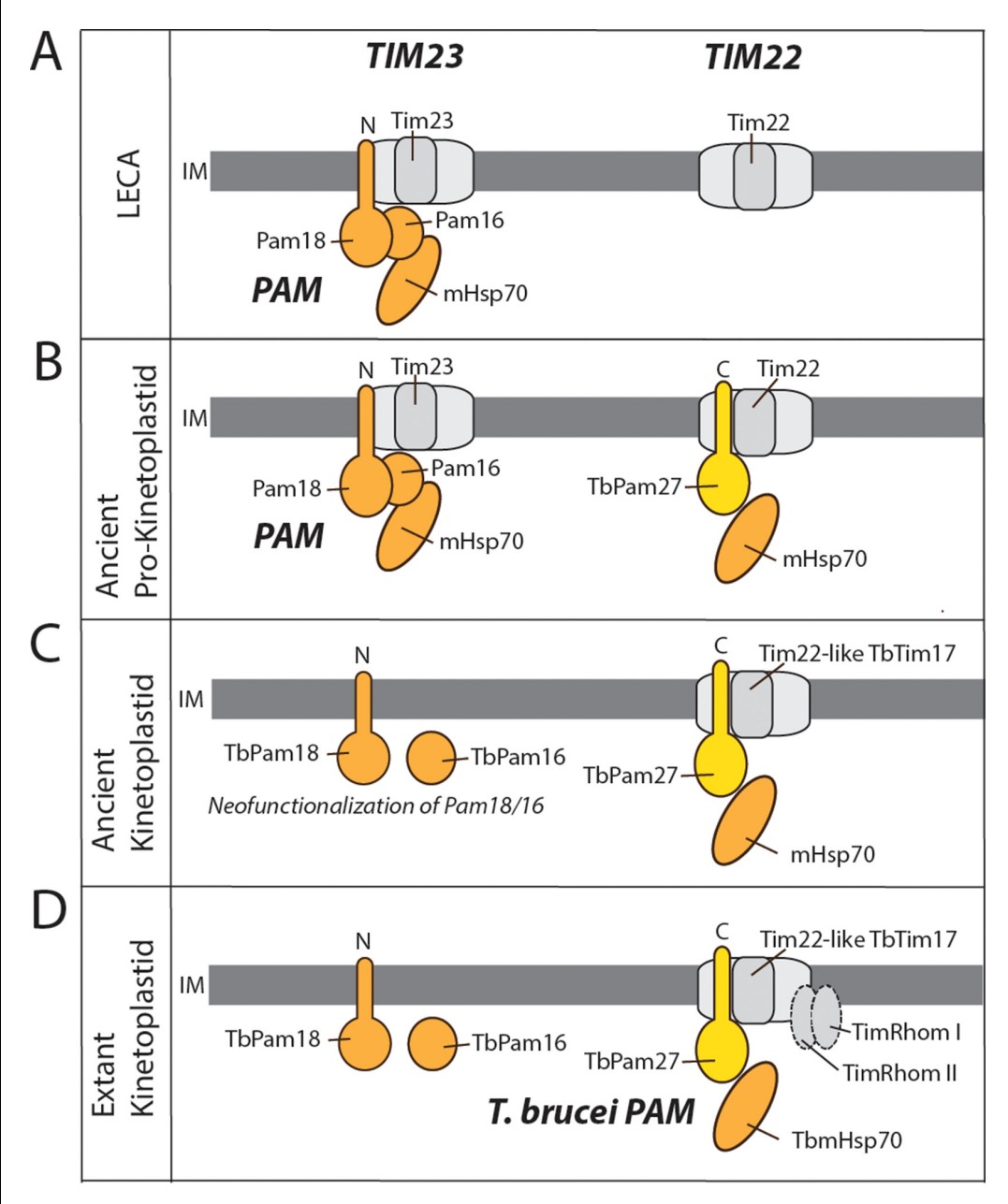

**Figure 10.** Evolutionary scenario explaining the homologue replacement observed in the *T. brucei* PAM. (**A**) Ancestral situation predicted for LECA. TIM23 and TIM22 complexes are shown in light gray their and respective pore-forming subunits (Tim23, Tim22) are indicated. PAM subunits are shown in dark orange. Only Pam18, Pam16 and mHsp70 are shown. (**B**) The J domain-containing protein TbPam27 (light orange) is recruited to the ancient pro-kinetoplastid TIM22 complex and may has enabled interaction with mHsp70. Initially this would have been neutral. (**C**) The TIM22 complex of the ancient kinetoplastid acquires the capability to translocate presequence-containing proteins, possibly through the recruitment of the trypanosomal presequence translocase specific TimRhom I and TimRhom II (**D**). This allows for deleterious mutations and eventual disappearance of the trypanosomal TIM23 complex. TbPam18 and TbPam16 were retained because they either acquired a new as yet unknown function, or hold an overlooked ancestral role.

ones seen in extant yeast and animals (*Figure 10A*). TbTim17, a *T. brucei* TIM core subunit, is most closely related to Tim22, which forms the pore of the TIM22 complex (*Žárský and Doležal, 2016*; *Pyrihová et al., 2018*). Thus, the transition from the ancestral state of two TIM complexes, to a

single bifunctional TIM complex, required modifications of the TIM22 complex that would allow for compensation of the loss of the TIM23 complex. This means that an ancestral version of the trypanosomal TIM22 complex acquired the ability to translocate presequence-containing proteins. A speculative, yet plausible, scenario for this conversion looks as follows: A J domain-containing protein with a C-terminal TMD, the ancestor of TbPam27, that possibly arose by gene duplication, was recruited to the mitochondrial IM. This protein by chance interacted with the TIM22 complex and facilitated its interaction with TbmHsp70 (*Figure 10B*). This modification would have originally been neutral but could have been the first step towards a bifunctional TIM22 complex since it may have buffered eventual deleterious mutations that led to the loss of the TIM23 complex (*Figure 10C*). TimRhom I and TimRhom II are specifically associated with the presequence translocase (*Harsman et al., 2016*). The recruitment of these proteins as auxiliary components that facilitate the import of presequence-containing proteins, may have further promoted the transition to a single TIM complex (*Figure 10D*).

Importantly, the fact that we have identified orthologues of Pam18 and Pam16, which do not retain their ancestral function, puts a wrench into comparative genomic investigations that assume retention of ancestral features means conservation of function. Moreover, our findings raise the question of why Pam16 and Pam18 are still maintained in *T. brucei* and other kinetoplastids, even though they are no longer necessary for the presequence import pathway. We see two possibilities to explain this. The first is simply by neo-functionalization. The proteins may have picked up a novel function in the kinetoplastid lineage during the course of the evolution of the bifunctional TIM complex. The second possibility is that TbPam16 and TbPam18 have retained an as yet unknown ancestral function that is required in all eukaryotes, but has not been identified due to the focus on their role in the PAM complex. It is currently unclear what the functions of TbPam18 and TbPam16 are. Interestingly, unlike its yeast homologue, TbPam18 seems to be dually localized in the ER and in mitochondria. Moreover, retention of both proteins in all kinetoplastids and the fact that their RNAi-mediated ablation in *T. brucei* specifically interferes with growth of the procyclic but not the bloodstream forms provides both, further information and constraints regarding their possible function. Investigation of TbPam16 and TbPam18 functions will be an interesting topic for future research.

In summary, our study illustrates the intricate and variable paths evolution may take in different phylogenetic groups to achieve the same aim, namely, to import and sort more than 1000 cytosolically-synthesized proteins into mitochondria. Our work underscores the necessity to combine bioinformatics with functional data to allow for a meaningful comparative analysis that can provide insight into eukaryotic diversification.

# Materials and methods

**Key resources table**

| Reagent type (species) or resource | Designation | Source or reference | Identifiers | Additional information |
|---|---|---|---|---|
| Gene (*Trypanosoma brucei*) | TbPam27 | | TriTrypDB: Tb927.10.13830 | |
| Gene (*T. brucei*) | TbPam18 | | TriTrypDB: Tb927.8.6310 | |
| Gene (*T. brucei*) | TbPam16 | | TriTrypDB: Tb927.9.13530 | |
| Gene (*T. brucei*) | Tb927.4.650 | | TriTrypDB: Tb927.4.650 | |
| Gene (*T. brucei*) | TbmHsp70 | | TriTrypDB: Tb927.6.3740; Tb927.6.3750; Tb927.6.3800 | |
| Gene (*T. brucei*) | ACAD | | TriTrypDB: Tb927.8.1420 | |
| Cell line (*T. brucei*) | 29–13, procyclic | PMID: 10215027 | | |

*Continued on next page*

*Continued*

| Reagent type (species) or resource | Designation | Source or reference | Identifiers | Additional information |
|---|---|---|---|---|
| Cell line (*T. brucei*) | L1γL262P, BSF | PMID: 23959897 | | |
| Antibody | Anti-HA (mouse, monoclonal) | BioLegend | 901503 (MMS-101R) | WB(1:5000) |
| Antibody | Anti-myc (mouse, monoclonal) | Invitrogen | 132500 | WB(1:2000) |
| Antibody | Anti-ATOM40 (rabbit, polyclonal) | Other | | Previously produced in our lab WB(1:10000) |
| Antibody | Anti-TbTim17 (rat, polyclonal) | Other | | Previously produced in our lab WB(1:100) |
| Antibody | Anti-TimRhom I (rabbit, polyclonal) | Other | | Previously produced in our lab WB(1:150) |
| Antibody | Anti-VDAC (rabbit, polyclonal) | Other | | Previously produced in our lab WB(1:1000) |
| Antibody | Anti-Cox IV (rabbit, polyclonal) | Other | | Previously produced in our lab WB(1:1000) |
| Antibody | Anti-EF1a (mouse, monoclonal) | Merk Millipore | 05–235 | WB(1:10000) |
| Antibody | Anti-Cytochrome C (rabbit, polyclonal) | Other | | Previously produced in our lab WB(1:100) |
| Antibody | Anti-ATOM69 (rabbit, polyclonal) | Other | | Previously produced in our lab WB(1:50) |
| Antibody | Anti-mouse (goat, HRP-coupled) | Sigma Aldrich | A4416 | WB(1:5000) |
| Antibody | Anti-mouse IRDye 680LT conjugated (goat) | LI-COR Biosciences | PN 926–68020 | WB(1:20000) |
| Antibody | Anti-rabbit IRDye 800CW conjugated (goat) | LI-COR Biosciences | PN 926–32211 | WB(1:20000) |
| Recombinant DNA reagent | TbPam27 ORF RNAi (plasmid) | This paper | | See Material and methods |
| Recombinant DNA reagent | TbPam27 5'UTR RNAi (plasmid) | This paper | | See Material and methods |
| Recombinant DNA reagent | TbPam18 ORF RNAi (plasmid) | This paper | | See Material and methods |
| Recombinant DNA reagent | TbPam16 ORF RNAi (plasmid) | This paper | | See Material and methods |
| Recombinant DNA reagent | Tb927.4.650 ORF RNAi (plasmid) | This paper | | See Material and methods |
| Recombinant DNA reagent | TbPam27-3xmyc (plasmid) | This paper | | See Material and methods |
| Recombinant DNA reagent | TbPam27-3xHA (plasmid) | This paper | | See Material and methods |
| Recombinant DNA reagent | TbPam27$^{wt}$ (plasmid) | This paper | | See Material and methods |
| Recombinant DNA reagent | TbPam27$^{H77Q}$ (plasmid) | This paper | | See Material and methods |
| Recombinant DNA reagent | TbPam18-3xHA (plasmid) | This paper | | See Material and methods |

*Continued on next page*

*Continued*

| Reagent type (species) or resource | Designation | Source or reference | Identifiers | Additional information |
|---|---|---|---|---|
| Recombinant DNA reagent | TbPam16-3xHA (plasmid) | This paper | | See Material and methods |
| Recombinant DNA reagent | Tb927.4.650-3xHA (plasmid) | This paper | | See Material and methods |
| Recombinant DNA reagent | LDH-DHFR-3xHA (plasmid) | PMID: 27991487 | | |
| Recombinant DNA reagent | LDH-DHFR-3xmyc (plasmid) | PMID: 27991487 | | |
| Recombinant DNA reagent | MCP12ΔI-myc (plasmid) | PMID: 27991487 | | |
| Commercial assay or kit | Prime-a-Gene labelling kit | Promega | U1100 | Radioactive labelling of Northern probes |
| Commercial assay or kit | SuperSignal West Femto maximum sensitivity substrate | Thermo Scientific | 34096 | Detection of BN-PAGE Western blot signals |
| Chemical compound, drug | Tetracycline Hydrochloride | Sigma Aldrich | T7660 | Induction of gene expression |
| Chemical compound, drug | Aminopterin | Sigma Aldrich | A1784 | Presequence pathway intermediate stalling |
| Chemical compound, drug | Sulfanilamide | Sigma Aldrich | S9251-100G | Presequence pathway intermediate stalling |
| Chemical compound, drug | Digitonin | Biosynth | 103203 | Generation of crude mitochondrial fractions |
| Chemical compound, drug | Lysine-L U-13C, U-15N | Euroisotop | CNLM-291-H-0.5 | SILAC labelling |
| Chemical compound, drug | Arginine-L U-13C6, U-15N4 | Euroisotop | CNLM-539-H-1 | SILAC labelling |
| Software, algorithm | GraphPad Prism, version 6.0 f | GraphPad Software | www.graphpad.com | Depiction of growth curves, quantifications and volcano blots. |
| Software, algorithm | ImageJ | | https://doi.org/10.1038/nmeth.2089 | Densitometric quantifications |
| Other | EZView Red Anti-c-myc affinity gel | Sigma Aldrich | E6654 | CoIP |
| Other | Anti-HA affinity matrix | Roche | 11815016001 | CoIP |

## Protein similarity network analyses

A database that comprised 46 proteomes from diverse representative eukaryotes, with an emphasis on euglenozoans (i.e., kinetoplastids, euglenoids, diplonemids), was assembled. HMMER searches (*Eddy, 1998*) using the Pfam profile Hidden Markov Models (pHMMs) for the DnaJ (PF00226) and Pam16 (PF03656) domains, and leniant E-values of 0.1 in order to capture divergent DnaJ domains, were performed against each proteome individually. The hit sequences were then used to build a local custom database of DnaJ- and Pam16 domain-containing proteins. Additionally, reciprocal BLASTp searches (*Altschul et al., 1990*) using ScPam16 and ScPam18 and the putative TbPam16 and TbPam18 as queries (identified through the network analysis) were performed. This was done to retrieve highly divergent proteins in some eukaryotes (e.g. in euglenozoans) that could not be captured from the curated Pfam profile HMMs. The reciprocal best BLAST hits retrieved were then added to the database of DnaJ and Pam16 domain-containing proteins built with pHMM searches. Duplicates were removed and the redundancy of the database was further reduced with CD-HIT and a threshold of 95% identity (*Li and Godzik, 2006*). An all-against-all BLAST search was performed

for this database using the BLOSUM50 matrix and an E-value of 0.1 as a threshold. The similarities (i. e., local alignments) among all DnaJ- and Pam16-domain containing proteins were visualized as a 2D protein similarity network with the software CLANS (*Frickey and Lupas, 2004*). Each DnaJ domain-containing protein (a node) is iteratively placed (for at least 1000 rounds) in a two-dimensional space according to attraction and repulsion values derived from BLAST P-values (edges). Edges correspond to P-values and nodes to protein sequences. Some isolated peripheral nodes and weakly intraconnected clusters were removed for a better visualization of the core similarity network.

## Assembly of diplonemid transcriptomes

Illumina short reads for RNA-Seq experiments of the diplonemids *Diplonema ambulator* (SRR5998378 and SRR5998379), *Diplonema* sp. (SRR5998375 and SRR5998376), and *Rhynchopus euleeides* (SRR5998382 and SRR5998383) were downloaded from NCBI SRA. Quality assessments of the RNA-Seq short raw reads were done with FastQC v0.11.7. The Illumina short reads were then quality-filtered with Trimmomatic v0.39 (*Bolger et al., 2014*). Quality filtered reads were assembled into transcriptomes with Trinity v2.8.4 (*Haas et al., 2013*). Proteomes was predicted from the assembled transcriptome with TransDecoder v5.5.0 (*Haas et al., 2013*).

## Phylogenetic analysis

Clusters in the similarity network corresponding to Pam16, Pam18, Pam27 and Tb927.4.650 protein families were retrieved and aligned to the Pfam Pam16 pHMM (*Eddy, 1998*). Non-aligned ends were trimmed and two extremely divergent sequences were also removed. Phylogenetic inference was performed under the maximum-likelihood framework using the software IQ-TREE (*Nguyen et al., 2015*). The best-fitting model was found to be LG+R5 by IQ-TREE's ModelFinder (*Kalyaanamoorthy et al., 2017*). Statistical branch support was assessed with Shimodaira-Hasegawa approximate Likelihood Ratio Test (SH-aLRT) and UltraFast Bootstrap two with NNI optimization (UFBoot2+NNI) (*Hoang et al., 2018*).

## Transgenic cell lines

Transgenic *T. brucei* cell lines were generated using the procyclic strain 29–13 (*Wirtz et al., 1999*) or the bloodstream form (BSF) strain New York single marker variant F1γL262P (*Dean et al., 2013*). Procyclic forms were cultivated at 27°C in SDM-79 (*Brun and Schönenberger, 1979*) supplemented with 10% (v/v) fetal calf serum (FCS). BSF cells were grown at 37°C in HMI-9 (*Hirumi and Hirumi, 1989*) containing 10% FCS (v/v).

To produce plasmids for ectopic expression of C-terminal triple c-myc- or HA-tagged TbPam27 (Tb927.10.13830), TbPam18 (Tb927.8.6310), TbPam16 (Tb927.9.13530), Tb927.4.650 and ACAD (Tb927.8.1420) the complete ORFs of the respective gene was amplified by PCR. The PCR products were cloned into a modified pLew100 vector (*Wirtz et al., 1999*; *Bochud-Allemann and Schneider, 2002*) which contains either a puromycin or a blasticidin resistance gene instead of phleomycin as well as a triple c-myc- or HA-tag (*Oberholzer et al., 2006*). The triple c-myc-tagged, truncated (nt 274–912) ORF of MCP12 (Tb927.10.12840) and both of the triple c-myc- or HA-tagged LDH-DHFR fusion proteins have been described previously (*Harsman et al., 2016*).

RNAi cell lines were prepared using the identical pLew100-derived vector described above. It allows the generation of a stem-loop construct by the insertion of the RNAi inserts in opposing directions. The loop is formed by a 460 bp spacer fragment. The RNAi targets the indicated nt of the ORFs of TbPam27 (nt 5–450), TbPam18 (nt 8–353), TbPam16 (nt 15–485) and Tb927.4.650 (nt 105–622) as well as the 5' untranslated region (UTR) of TbPam27 (nt (−422)-(−66)). RNAi against the ORF of TbmHsp70 (Tb927.6.3740, Tb927.6.3750, Tb927.6.3800) has been described previously (*Tschopp et al., 2011*).

The H77Q mutant of TbPam27 (TbPam$^{H77Q}$) was constructed by site-directed mutagenesis using complementary primers that carry the desired mutation, as well as a down- and an upstream primer. PCR products of the wildtype and the H77Q mutant version of the ORF of TbPam27 were cloned into a pLew100 derived vector lacking the epitope tag.

### Digitonin extraction

Crude mitochondria-enriched fractions were obtained by incubating $1 \times 10^8$ cells on ice for 10 min in 0.6 M sorbitol, 20 mM Tris-HCl pH 7.5, 2 mM EDTA pH 8 containing 0.015% (w/v) digitonin for selective solubilization of plasma membranes. Centrifugation (5 min, 6'800 g, 4°C) yielded a supernatant that is enriched in cytosolic proteins and a mitochondria-enriched pellet. Equivalents of $2 \times 10^6$ cells of each fraction were analyzed by SDS-PAGE and subsequent Western blotting to demonstrate mitochondrial localization for proteins of interest. Cell lines were induced with tetracycline (Tet) for 24 hr to visualize epitope-tagged proteins.

### Alkaline carbonate extraction

In order to separate soluble or loosely membrane associated proteins from integral membrane proteins, a mitochondria-enriched pellet was generated as described above and resuspended in 100 mM $Na_2CO_3$ pH 11.5, incubated on ice for 10 min and centrifuged (10 min, 100'000 g, 4°C). Equivalents of $2 \times 10^6$ cells of each fraction were subjected to SDS-PAGE and Western blotting.

### Protease protection assay

A mitochondria-enriched digitonin pellet from $1 \times 10^7$ cells per sample was generated as described above. This pellet was resuspended in 250 mM sucrose, 80 mM KCl, 5 mM MgAc, 2 mM $KH_2PO_4$ and 50 mM HEPES in a total volume of 50 μl and the indicated additions of proteinase K and 0.5% (v/v) Triton-X100. After 15 min incubation on ice, the reactions were stopped by adding PMSF to 5 mM. Samples without Triton X100 were centrifuged (5 min, 6800 g, 4°C) and all samples were resuspended in SDS loading buffer. Of each sample, $1 \times 10^6$ cell equivalents were subjected to SDS-PAGE and Western blotting.

### Co-immunoprecipitation

Digitonin-extracted mitochondria-enriched fractions of $1 \times 10^8$ cells expressing the epitope-tagged protein of interest were solubilized for 15 min on ice in 20 mM Tris-HCl pH7.4, 0.1 mM EDTA, 100 mM NaCl, 10% glycerol containing 1X Protease Inhibitor mix (Roche, EDTA-free) and 1% (w/v) digitonin. After centrifugation (15 min, 20'000 g, 4°C), the lysate (input) was transferred to either 50 μl of HA bead slurry (anti-HA affinity matrix, Roche) or 30 μl c-myc bead slurry (EZview red anti-c-myc affinity gel, Sigma) both of which had been equilibrated in wash buffer (20 mM Tris-HCl pH 7.4, 0.1 mM EDTA, 100 mM NaCl, 10% glycerol, 0.2% (w/v) digitonin). After incubating at 4°C for 1 hr, the supernatant containing the unbound proteins was removed. The bead slurry was washed three times with wash buffer and the bound proteins were eluted by boiling the resin for 5 min in 2% SDS in 60 mM Tris-HCl pH 6.8 (IP). Five percent of both the input and the unbound proteins and 100% of the IP sample were subjected to SDS-PAGE and Western blotting.

For the stalled import intermediate, LDH-DHFR-HA expression was induced by tetracycline and respective cell cultures were supplemented with 1 mM sulfanilamide and 50 μM aminopterine (AMT) 12 hr before the experiment (*Harsman et al., 2016*). CoIP was performed as described above.

### Blue native-PAGE

Digitonin-extracted mitochondria-enriched fractions were incubated for 15 min on ice in 20 mM Tris-HCl pH 7.4, 50 mM NaCl, 10% glycerol, 0.1 mM EDTA, 1 mM PMSF containing 1% (w/v) digitonin in order to solubilize mitochondrial membranes. After a clearing spin (15 min, 20'817 g, 4°C) the samples were separated on 4–13% gradient gels. Prior to Western blotting the gel was incubated in SDS-PAGE running buffer (25 mM Tris, 1 mM EDTA, 190 mM glycine, 0.05% (w/v) SDS) in order to facilitate the transfer of proteins to the membrane.

### RNA extraction and northern blotting

Acid guanidinium thiocyanate-phenol-chloroform extraction according to *Chomczynski and Sacchi (1987)* was used for isolation of total RNA from uninduced and induced (2 days) RNAi cells. RNA samples were separated on a 1% agarose gel in 20 mM MOPS buffer supplemented with 0.5% formaldehyde. Northern probes were generated from gel-purified PCR products corresponding to the RNAi inserts mentioned above and radioactively labelled by means of the Prime-a-Gene labelling system (Promega).

## SILAC proteomics and IP

TbPam27 ORF RNAi cells or cells that allow inducible expression of epitope-tagged proteins were washed in PBS and taken up in SDM-80 (*Lamour et al., 2005*) supplemented with 5.55 mM glucose, either light ($^{12}C_6/^{14}N_\chi$) or heavy ($^{13}C_6/^{15}N_\chi$) isotopes of arginine (1.1 mM) and lysine (0.4 mM) (Euroisotop) and 10% dialyzed FCS (BioConcept, Switzerland). To guarantee complete labeling of all proteins with heavy amino acids, the cells were cultured in SILAC medium for 6–10 doubling times.

For the TbPam27 SILAC RNAi experiment, the cells were induced with tetracycline for 1.5 days. Directly before cell lysis, uninduced and induced cells were mixed in a 1:1 ratio. Of the mixed cells, digitonin-extracted mitochondria-enriched pellets were generated.

For the ACAD, TbPam18 and TbPam16 SILAC-IP experiments, digitonin-extracted mitochondria-enriched pellets of $2 \times 10^8$ uninduced and induced cells were subjected to CoIP as described above.

The TbPam27 SILAC RNAi experiment and all SILAC-IPs were done in three biological replicates including a label-switch and analyzed by liquid chromatography-mass spectrometry (LC-MS).

## LC-MS and data analysis

Proteins of mitochondrial fractions prepared from SILAC-labeled induced and uninduced TbPam27-RNAi cells (20 µg per replicate; n = 3) were reduced, alkylated, and tryptically digested as described before (*Peikert et al., 2017*). To reduce sample complexity, peptides were fractionated by high-pH reversed-phase chromatography using StageTips (*Rappsilber et al., 2007*). Peptides were dried *in vacuo*, resuspended in 10 mM NH$_4$OH and loaded onto StageTips that were equilibrated with 10 mM NH$_4$OH. A step gradient of 0, 2.7, 5.4, 9.0, 11.7, 14.4, 36% and 65% (v/v) acetonitrile, diluted in 10 mM NH$_4$OH, was applied for peptide elution. Proteins present in eluates of ACAD, TbPam16 and TbPam18 SILAC-IPs (n = 3 each) were processed following a gel-based approach. Reduction, alkylation and tryptic in-gel digestion of proteins were performed as described (*Peikert et al., 2017*) and the resulting peptide mixtures were desalted using StageTips. Peptides eluted from StageTips were dried *in vacuo*, reconstituted in 0.1% trifluoroacetic acid, and analyzed by LC-MS using a Q Exactive Plus (samples of TbPam27-RNAi experiments and TbPam16 and TbPam18 SILAC-IPs) or an Orbitrap Elite (ACAD SILAC-IPs) (Thermo Fisher Scientific, Germany) coupled to an UltiMate 3000 RSLCnano HPLC system (Thermo Fisher Scientific, Germany). For the elution of peptides, the following solvent systems were used: 0.1% (v/v) formic acid (FA) (solvent A) and 86% (v/v) acetonitrile (ACN)/0.1% (v/v) FA (solvent B) for LC-MS analyses using the Q Exactive and 0.1% (v/v) FA/4% (v/v) DMSO (solvent A') and 50% (v/v) methanol/30% (v/v) ACN/0.1% (v/v) FA/4% (v/v) DMSO (solvent B') for analyses using the Orbitrap Elite. Peptides generated in TbPam27-RNAi experiments were eluted with a solvent gradient increasing from 4–40% B in 50 min followed by 40–95% B in 5 min and 5 min at 95% B. Peptides of TbPam16 and TbPam18 complexes were eluted with 1–20% B in 103 min, 20–42% B in 50 min, 42–95% B in 2 min and 5 min at 95% B, and for the elution of peptides from ACAD SILAC-IPs, a gradient ranging from 1–65% B' in 30 min followed by 65–95% B' in 5 min and 5 min at 95% B' was applied. The flow rates were 300 nl/min.

The Q Exactive Plus instrument was operated with the following mass spectrometric parameters: MS scan range, *m/z* 375–1,700; resolution, 70,000 (at *m/z* 200); target value, $3 \times 10^6$ ions; max injection time, 60 ms; TOP12-higher-energy collisional dissociation of multiply charged peptides; dynamic exclusion time, 45 s. For MS analyses at the Orbitrap Elite, the settings were as follows: MS scan range, *m/z* 370–1,700; resolution, 120,000 (at *m/z* 400); target value, $1 \times 10^6$ ions; max injection time, 200 ms; TOP5-collision-induced dissociation of multiply charged peptides; dynamic exclusion time, 45 s.

MaxQuant/Andromeda (version 1.5.4.0 for ACAD SILAC-IPs and 1.5.5.1 for the other samples [*Cox et al., 2011*; *Cox and Mann, 2008*] was used for protein identification and relative quantification. MS/MS data were searched against all entries for *T. brucei* TREU927 retrieved from the TriTryp database (version 8.1) using MaxQuant default settings except that one unique peptide was sufficient for protein identification. Lys8 and Arg10 were selected as heavy labels, carbamidomethylation of cysteine as fixed, and N-terminal acetylation and oxidation of methionine as variable modifications. Protein abundance ratios were calculated based on unique peptides and at least one SILAC peptide pair. Lists of proteins identified and quantified in the analysis of TbPam27-RNAi experiments as well as SILAC-IPs of ACAD, TbPam16 and TbPam18 are provided in *Figure 3—source datas 1–3* and *Figure 9—source data 1*.

## Additional information

### Funding

| Funder | Grant reference number | Author |
|---|---|---|
| Swiss National Science Foundation | 175563 | Andre Schneider |
| Swiss National Science Foundation | NCCR RNA and Disease | Andre Schneider |
| European Research Council | Consolidator grant 648235 | Bettina Warscheid |
| Deutsche Forschungsgemeinschaft | 403222702/SFB 1381 | Bettina Warscheid |
| Germany's Excellence Strategy | CIBSS - EXC-2189 - Project ID 390939984 | Bettina Warscheid |
| Excellence Initiative of the German Federal and State Governments | EXC 294 BIOSS | Bettina Warscheid |
| Peter und Traudl Engelhorn foundation | | Anke Harsman |

The funders had no role in study design, data collection and interpretation, or the decision to submit the work for publication.

### Author contributions

Corinne von Känel, Conceptualization, Formal analysis, Validation, Investigation, Visualization, Methodology, Writing - original draft, Writing - review and editing; Sergio A Muñoz-Gómez, Resources, Formal analysis, Visualization, Methodology, Writing - review and editing; Silke Oeljeklaus, Resources, Formal analysis, Methodology, Writing - review and editing; Christoph Wenger, Investigation, Writing - review and editing; Bettina Warscheid, Resources, Funding acquisition, Project administration, Writing - review and editing; Jeremy G Wideman, Conceptualization, Investigation, Visualization, Writing - original draft, Writing - review and editing; Anke Harsman, Conceptualization, Formal analysis, Supervision, Validation, Investigation, Methodology, Writing - review and editing; Andre Schneider, Conceptualization, Formal analysis, Supervision, Funding acquisition, Visualization, Writing - original draft, Project administration, Writing - review and editing

### Author ORCIDs

Corinne von Känel (iD) https://orcid.org/0000-0002-2885-5161
Bettina Warscheid (iD) http://orcid.org/0000-0001-5096-1975
Andre Schneider (iD) https://orcid.org/0000-0001-5421-0909

### Decision letter and Author response

Decision letter https://doi.org/10.7554/eLife.52560.sa1
Author response https://doi.org/10.7554/eLife.52560.sa2

# Additional files

### Supplementary files

• Transparent reporting form

### Data availability

All produced data are contained within the manuscript (e.g. Data Source files).

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
