## [Decision Letter]

**Acceptance summary:**

The mitochondrial protein import machinery of Trypanosomes differs from that of fungi and animals. In contrast to fungi and animals, only one TIM machinery is used by the proteins that translocate to the mitochondrial matrix and the proteins, such as carriers, that integrate into the inner mitochondrial membrane. Interestingly, the study shows that a novel protein TbPam27 takes the role of presequence translocase associated motor (PAM) and its J protein, Pam18, in the TIM complex serving an important role during the protein import into mitochondria. The study provides an interesting example of the functional and architectural replacement of proteins during the transition into bifunctional TIM complex in the evolution.

**Decision letter after peer review:**

Thank you for submitting your article "Homologue replacement in the import motor of the mitochondrial inner membrane of trypanosomes" for consideration by *eLife*. Your article has been reviewed by three peer reviewers, and the evaluation has been overseen by a Reviewing Editor and Dominique Soldati-Favre as the Senior Editor. The following individual involved in review of your submission has agreed to reveal their identity: Patrick R D' Silva (Reviewer #3).

The reviewers have discussed the reviews with one another and the Reviewing Editor has drafted this decision to help you prepare a revised submission.

The reviewers are overall positive, however they identified several issues that must be addressed in the revision of this work.

All of them agree that the this work does not sufficiently demonstrate the interactions as well as the role Pam27 in relation to other Pam proteins at the TbTIM complex.

Concerning the architecture of TIM and Pam27 specifically, the authors should examine immunoprecipitation experiments for other known components of the TIM complex, either via antibodies and/or mass spectrometry analyses. The reverse immunoprecipitations should also be performed, also using Pam18 and Pam16 as baits. All the experiments addressing the assemblies of the TbTIM complex should be carefully controlled including negative controls.

The reviewers are not convinced that a single entity ThTIM complex exists. Another possibility is that they are several sub-pools of this complexes, some of them without Pam27. The authors should consider some of the following experiments to address this issue: testing whether TbPam18 and TbPam16 are recruited to TIM-complex in the absence of TbPam27, testing the levels of TbPam18 and TbPam16 in TbPam27 RNAi cells. An interesting input can also be obtained from BN analyses.

A functional aspect of the study must be strengthened in general, and also to provide more insights into functionally identical vs. diverse complex assemblies. TbPam16, TbPam18 and TbPam27 are all essential proteins. However, the roles of TbPam16 and TbPam18 for protein import unclear. The authors should elucidate this in more depth, by utilizing import assays, and also for example by mass spectrometry of cells in which these components were depleted.

Other specific points of reviewers should or may be considered prior to submission of the revised version:

Reviewer 1:

1) Figure 2 – evidence for inner membrane localisation is weak and indirect. This should be done using a mitochondrial subfractionation assay, probing with and without protease.

2) Figure 4 uses MCP1Δ as a control for LDH-DHFR, however given this is control precursor is not fused to DHFR it is not an appropriate control? Given this, the outcome of this figure, i.e. carrier biogenesis is not affected is not convincing enough. The authors should also show both OM and IM IP controls for DHFR blocking as spans both membrane.

3) Figure 5, Blots in A (right panel) are inconsistent with the time frame of graph (missing day 4) and remainder of blots shown in figure.

4) Figure 7, need complementary SDS PAGE analysis to show levels of Myc protein and levels of TIM subunits. In panel B the authors suggest the carrier intermediate is not affected (in the text), but it clearly is when looking at the data. This data should not be misinterpreted and the possibility that carrier import is also affected should be considered.

Figure legend and description of which samples go into (C) is not clear. Perhaps also state in axis label what the band intensity was calculated from.

Top left: what does the 34.4% correspond too? Not mentioned in main text or legend.

Top right: label some of most significant presequence-containing proteins.

Bottom left: how is the 8.3% calculated? Poorly described in text/not at all in legend.

Bottom right: make bottom left (and give letters) as referred to first in text.

Reviewer 2:

Figure 8 contains little information and should be shifted into the supplements. Instead, the mass spec data shown in Figure 9 should be further analyzed. For example, measurements of the proteomes after short or longer times of ablation might allow to distinguish direct from indirect effects. In particular proteins would be interesting, which disappear very early after downregulation of TbPam27.*Reviewer 3:*

1) VDAC is recently shown to interact with yeast TIM22-complex for the efficient import of Carrier proteins (https://doi.org/10.1016/j.molcel.2018.12.014); therefore, the observed pull-down of TbTim17 and VDAC with Tb927.4.650-HA might be specific (although having lesser affinity), thereby Tb927.4.650 could be a subunit of Tb TIM complex required for carrier protein import. Similarly, VDAC might be cooperating with the TbTIM complex during carrier protein import.

---

## [Author Response]

The reviewers are overall positive, however they identified several issues that must be addressed in the revision of this work.All of them agree that the this work does not sufficiently demonstrate the interactions as well as the role Pam27 in relation to other Pam proteins at the TbTIM complex.Concerning the architecture of TIM and Pam27 specifically, the authors should examine immunoprecipitation experiments for other known components of the TIM complex, either via antibodies and/or mass spectrometry analyses.

The composition of the trypanosomal TIM complex has been extensively analyzed by proteomic analysis of SILAC immunoprecipitations (IP) using the tagged TIM subunits TbTim17, TbTim42 and TbTim13 (a TIM complex-associated small TIM chaperone) as baits (1). Moreover, the same analysis was also done using a stalled presequence-containing precursor protein or a stalled mitochondrial carrier as a bait (1). In the revised manuscript we have now complemented this data by a new SILAC-IP, which uses the tagged TIM subunit ACAD as a bait (new Figure 3—source data 1). A summary of the six reciprocal IPs is provided in the table shown in "Figure 3—figure supplement 1" of the revised manuscript. It shows that TbPam27 is enriched in 5 out of 6 IPs, whereas TbPam18, TbPam16 and Tb927.4.650 are either not detected or not significantly enriched in any of the 6 IPs (compared to VDAC which is used as a control).

The new analysis is discussed in the revised manuscript (subsection "TbPam27, but not TbPam18, TbPam16 or Tb927.4.650, is associated with the TIM complex", second paragraph and subsection "TbPam27 is part of the active presequence translocase", third paragraph).

The reverse immunoprecipitations should also be performed, also using Pam18 and Pam16 as baits. All the experiments addressing the assemblies of the TbTIM complex should be carefully controlled including negative controls.

We preformed the reverse SILAC proteomics-based IPs using C-terminally tagged TbPam18 and TbPam16 as baits, as suggested (new Figure 3B in the revised manuscript). Again, the results were very clear, none of the detected TIM subunits in these experiments was significantly enriched in either of the two IPs. This indicates that — in contrast to TbPam27 — neither TbPam18 nor TbPam16 interacts with the TIM complex. — Further results we gained from these experiments were: i) that TbPam16 essentially exclusively interacts with TbPam18, and that ii) TbPam18 likely has a dual localization in the mitochondrion and the ER. TbPam16 and TbPam18 have predicted N-terminal mitochondrial targeting sequences, it is therefore unlikely that the C-terminal tag interferes with their localization.

The new results are discussed in the revised manuscript (subsection "TbPam27, but not TbPam18, TbPam16 or Tb927.4.650, is associated with the TIM complex").

The reviewers are not convinced that a single entity ThTIM complex exists. Another possibility is that they are several sub-pools of this complexes, some of them without Pam27.

This is a misunderstanding, we don't claim that there is only a single entity of the TIM complex. In fact the pulldown experiments using the two different arrested import substrates show that TbPam27 is preferentially associated with the precursor and not the carrier translocase. — What we claim is that there is no evidence that either TbPam18 or TbPam16 is associated with any form of the TIM complex that we can detect in six different SILAC-IPs (see new Figure 3—figure supplement 1 and new Figure 3B in the revised manuscript).

The authors should consider some of the following experiments to address this issue: testing whether TbPam18 and TbPam16 are recruited to TIM-complex in the absence of TbPam27, testing the levels of TbPam18 and TbPam16 in TbPam27 RNAi cells. An interesting input can also be obtained from BN analyses.

We did the requested experiment (new Figure 6—figure supplement 1 of the revised manuscript). The results show i) that there is no compensatory accumulation of TbPam16 or TbPam18 upon ablation of TbPam27 and ii) that pull down experiments in TbPam27 ablated cells using tagged versions of TbPam16 or TbPam18 still do not recover the TIM subunits TbTim17 or TimRhom I.

The new results are discussed in revised manuscript (subsection "Neither TbPam16 nor TbPam18 is required for mitochondrial protein import", last paragraph).

We tried to do BN-PAGE analysis, however, the tagged TbPam18 or TbPam16 could not be detected in this type of analysis.

A functional aspect of the study must be strengthened in general, and also to provide more insights into functionally identical vs. diverse complex assemblies. TbPam16, TbPam18 and TbPam27 are all essential proteins. However, the roles of TbPam16 and TbPam18 for protein import unclear. The authors should elucidate this in more depth, by utilizing import assays, and also for example by mass spectrometry of cells in which these components were depleted.

*in vitro* import assays are feasible with mitochondria from wild-type *T. brucei*, but they are much more difficult with RNAi cells. For some cells linesthey do not work at all, even when using mitochondria from uninduced cells, which should not show a phenotype. Moreover, we only have a single substrate that works efficiently. The information that can be gained from such *in vitro* import experiments is therefore limited.

SILAC MS analysis of TbPam18 and TbPam16 RNAi cells could be done. However, the manuscript is about TbPam27 and we don't claim to know what the function of TbPam18 and TbPam16 is, except that they are not involved in import of presequence-containing proteins.

Instead we decided to do another experiment, which provides important information regarding the function of TbPam16 and TbPam18. *T. brucei* has a complex life cycle, alternating between an insect and a mammalian host. Mitochondrial protein import is essential for both insect stage and bloodstream forms of the parasite. We now show that only TbPam27, but neither TbPam16 nor TbPam18, is essential for growth of the bloodstream form (new Figure 5C and new Figure 6B of the revised manuscript). These results are consistent with the role of TbPam27 as a PAM subunit in mitochondrial protein import. Moreover, they show that the essential function of TbPam16 and TbPam18 is restricted to the insect stage of the parasite and, therefore, cannot be mitochondrial protein import.

The new results are discussed in the revised manuscript (subsection "TbPam27 with an intact J domain is essential for mitochondrial protein import", second paragraph and subsection "Neither TbPam16 nor TbPam18 is required for mitochondrial protein import", second paragraph).

Other specific points of reviewers should or may be considered prior to submission of the revised version:Reviewer 1:1) Figure 2 — evidence for inner membrane localisation is weak and indirect. This should be done using a mitochondrial subfractionation assay, probing with and without protease.

Mitochondrial subfractionation assays are possible with large amounts of cells on the preparative scale using sucrose gradient centrifugation (this data is provided for TbPam27 in Figure 2D of the revised manuscript) but they do not work on the analytical scale. — However, we now show that TbPam27, TbPam18 and TbPam16 are protease-resistant in digitonin-purified crude mitochondrial fractions indicating that all three proteins are localized in the mitochondrial inner and not the outer membrane. — It should be considered in this context that for TbPam18 the interpretation of the result is more complex, since it is likely dually localized to the mitochondrion and the ER.

The new results are discussed in the revised manuscript (subsection "TbPam27, but not TbPam18, TbPam16 or Tb927.4.650, is associated with the TIM complex", first paragraph).

2) Figure 4 uses MCP1Δ as a control for LDH-DHFR, however given this is control precursor is not fused to DHFR it is not an appropriate control? Given this, the outcome of this figure, i.e. carrier biogenesis is not affected is not convincing enough. The authors should also show both OM and IM IP controls for DHFR blocking as spans both membrane.

We are not sure we understand the reviewers concern. The two import intermediates have been described in detail in a previous publication (1). There it was also shown that the stalled presequence substrate spans both membranes, whereas the truncated carrier intermediate accumulates in the inner membrane only. Fusing the truncated carrier substrate to DHFR would block both the presequence and the carrier pathway.

3) Figure 5, Blots in A (right panel) are inconsistent with the time frame of graph (missing day 4) and remainder of blots shown in figure.

The missing data for day 4 has been added

4) Figure 7, need complementary SDS PAGE analysis to show levels of Myc protein and levels of TIM subunits. In panel B the authors suggest the carrier intermediate is not affected (in the text), but it clearly is when looking at the data. This data should not be misinterpreted and the possibility that carrier import is also affected should be considered.

The complementary SDS PAGE analysis has been added (Figure 7 of the revised manuscript). It monitors the levels of the TIM complex core subunit TbTim17 as well as the levels of LDH-DHFR and the MCP12ΔI substrates, respectively. (Note that for the LDH-DHFR substrate addition of aminopterin results in shift of the band on the SDS gel).

It is stated in the text that "…accumulation of the truncated MCP12ΔI was not affected for at least two days". This is indeed what Figure 7B shows. — We do see an effect on the carrier intermediate, but only after three days (and even then it is much less than in the case of the presequence intermediate), but these are likely pleiotropic effects caused by the growth arrest, which starts after day one. — It is important to note that eventually import of all proteins will be affected since import of some TIM core subunits (e.g. ACAD), which are required for import of both presequence-containing as well as mitochondrial carrier proteins, have predicted mitochondrial targeting sequences.

Figure legend and description of which samples go into (C) is not clear. Perhaps also state in axis label what the band intensity was calculated from.

The legend has been modified

Top left: what does the 34.4% correspond too? Not mentioned in main text or legend.

It is now mentioned in the revised manuscript (subsection "Ablation of TbPam27 preferentially affects presequence-containing proteins", first paragraph)

Top right: label some of most significant presequence-containing proteins.

Some proteins have been labeled in the revised figure.

Bottom left: how is the 8.3% calculated? Poorly described in text/not at all in legend.

The sentence in question (subsection "Ablation of TbPam27 preferentially affects presequence-containing proteins", first paragraph) has been modified.

Bottom right: make bottom left (and give letters) as referred to first in text.

The figure has been rearranged and re-labeled

Reviewer 2:Figure 8 contains little information and should be shifted into the supplements.

The reviewers put a lot of emphasis on TbPam18 and TbPam16 we would, therefore, like to keep it as a main figure.

Instead, the mass spec data shown in Figure 9 should be further analyzed. For example, measurements of the proteomes after short or longer times of ablation might allow to distinguish direct from indirect effects. In particular proteins would be interesting, which disappear very early after downregulation of TbPam27.

SILAC proteomics experiments are costly, which is why we analyzed a single time point only. However, we believe that the extent to which proteins get downregulated after ablation of TbPam27, can serve as a proxy for a time course. Thus, we hypothesize that the more efficiently downregulated proteins (we use a cutoff of 1.5-fold) are directly affected by the lack of TbPam27. Having said that, it is important to realize that these are complex *in vivo* experiments and there are always parameters that cannot be controlled, which may confound the results for a few proteins (e.g. the degree of downregulation is influenced by the half-life of the protein in question). The power of the analysis comes from the fact that we have data for 899 different proteins (83% of the mitochondrial proteome).

Reviewer 3:1) VDAC is recently shown to interact with yeast TIM22-complex for the efficient import of Carrier proteins (https://doi.org/10.1016/j.molcel.2018.12.014); therefore, the observed pull-down of TbTim17 and VDAC with Tb927.4.650-HA might be specific (although having lesser affinity), thereby Tb927.4.650 could be a subunit of Tb TIM complex required for carrier protein import. Similarly, VDAC might be cooperating with the TbTIM complex during carrier protein import.

The amount of TbTim17 and VDAC detected in the Tb927.4.650 pull downs is approximately the same as in the other tested cell lines, which is why we considered it not to be specific. It is true that we cannot exclude a low affinity association of the two proteins with Tb927.4.650. However, we did not further investigate Tb927.4.650 in the present study because, its ablation does not cause a growth arrest.